# Long-term soil warming decreases microbial phosphorus utilization by increasing abiotic phosphorus sorption and phosphorus losses

Ye Tian [1,2] ✉, Chupei Shi[1,3], Carolina Urbina Malo[1,4], Steve Kwatcho Kengdo [5], Jakob Heinzle[6], Erich Inselsbacher[7], Franz Ottner[8], Werner Borken[5], Kerstin Michel [6], Andreas Schindlbacher[6] & Wolfgang Wanek [1] ✉

Phosphorus (P) is an essential and often limiting element that could play a crucial role in terrestrial ecosystem responses to climate warming. However, it has yet remained unclear how different P cycling processes are affected by warming. Here we investigate the response of soil P pools and P cycling processes in a mountain forest after 14 years of soil warming (+4 °C). Long-term warming decreased soil total P pools, likely due to higher outputs of P from soils by increasing net plant P uptake and downward transportation of colloidal and particulate P. Warming increased the sorption strength to more recalcitrant soil P fractions (absorbed to iron oxyhydroxides and clays), thereby further reducing bioavailable P in soil solution. As a response, soil microbes enhanced the production of acid phosphatase, though this was not sufficient to avoid decreases of soil bioavailable P and microbial biomass P (and biotic phosphate immobilization). This study therefore highlights how long-term soil warming triggers changes in biotic and abiotic soil P pools and processes, which can potentially aggravate the P constraints of the trees and soil microbes and thereby negatively affect the C sequestration potential of these forests.

By the end of this century, global atmospheric temperatures are predicted to increase up to 5.7 °C above pre-industrial levels[1]. Elevated soil temperatures will substantially affect soil biogeochemical processes and element cycling, potentially feeding back on climate change itself[2,3]. In recent years, warming effects on soil phosphorus (P) have received increasing attention[4–6]. This is because P is vital for the growth, functioning, and reproduction of all organisms, thereby affecting ecosystem responses to global warming[7,8]. Until now, most P-centered warming studies in terrestrial ecosystems have focused on changes in plant P pools and their stoichiometric ratios with carbon (C) and nitrogen (N)[4,5,9], showing overall increases in plant N:P ratios[4]. Soil and plant P processes and P pools are highly connected and

[1]Division of Terrestrial Ecosystem Research, Department of Microbiology and Ecosystem Science, Center of Microbiology and Environmental Systems Science, University of Vienna, Djerassiplatz 1, A-1030 Vienna, Austria. [2]University of Vienna, Doctoral School in Microbiology and Environmental Science, Djerassiplatz 1, A-1030 Vienna, Austria. [3]Department of Ecosystem and Landscape Dynamics, Institute for Biodiversity and Ecosystem Dynamics, University of Amsterdam, Science Park 904, 1098 XH Amsterdam, Netherlands. [4]Institute of Soil Science, Leibniz Universität Hannover, Herrenhäuser Straße 2, 30419 Hannover, Germany. [5]Department of Soil Ecology, Bayreuth Center of Ecology and Environmental Research (BAYCEER), University of Bayreuth, Dr. Hans-Frisch-Straße 1-3, 95448 Bayreuth, Germany. [6]Department of Forest Ecology and Soil, Federal Research and Training Centre for Forests, Natural Hazards and Landscape-BFW, Seckendorff-Gudent Weg 8, 1130 Vienna, Austria. [7]Institute of Soil Research, Department of Forest and Soil Sciences, University of Natural Resources and Life Sciences Vienna (BOKU), Peter Jordan-Straße 82, 1190 Vienna, Austria. [8]Institute of Applied Geology, Department of Civil Engineering and Natural Hazards, University of Natural Resources and Life Sciences, University of Natural Resources and Life Sciences Vienna (BOKU), Peter Jordan-Straße 82, 1190 Vienna, Austria. ✉e-mail: ye.tian@univie.ac.at; wolfgang.wanek@univie.ac.at

interacting[8,10–13]. However, how specific P cycling processes respond to elevated temperatures has largely remained elusive, though these responses could critically affect plant P nutrition, as well as microbial soil C and N cycling[8,9].

Soil P cycling is driven by biotic and abiotic processes. Some studies have analyzed the biotic side by investigating warming effects on soil phosphatase activities and on soil organic P mineralization[6,14]. Globally, warming did not increase phosphatase activity[6], likely as a stimulative warming effect was offset due to concurrently reduced soil water availability[13,15]. On the other hand, the mineralization of organic P was stimulated[14] and soil P mobilization increased with warming[16], accompanied by increased plant P demand[17] and uptake[16]. Abiotic processes, i.e., adsorption/desorption and precipitation/dissolution processes, affect P availability in soil solution, adding another component to the complexity of soil P cycling. Adsorption is a thermodynamically driven process and phosphate sorption to dolomite has been shown to be an exothermic reaction[18–20]. Thus, increasing temperatures are expected to shift the equilibrium to a state of less adsorbed P, but this was not demonstrated yet. Given the lack of comprehensive understanding of warming effects on biotic and abiotic soil P processes, we here applied a $^{33}P$ isotope pool dilution method[21], which allows to quantify the process rates of gross and net (biotic and abiotic) P mobilization (reflecting microbial P mineralization as biotic process, and P desorption and dissolution as abiotic processes) and P immobilization (reflecting microbial P immobilization, and P sorption and precipitation). This allowed us to directly examine how specific soil P cycling processes responded to long-term soil warming in a temperate forest.

Our aim therefore was to investigate how soil warming affects soil P pools and soil P cycling processes. In a long-term (>14 years) forest soil warming experiment (+4 °C, during the snow-free seasons) situated in a temperate moist mountain forest (the Achenkirch soil warming experiment[22]), we tested the hypotheses that long-term warming will (i) promote biological soil P processes, i.e., phosphomonoesterase activity due to the lack of drought offsets of the warming effect, (ii) hinder P sorption relative to desorption, and (iii) thereby increase gross rates of soil P mobilization through biotic and abiotic processes. The increase in soil P mobilization will be reflected in increased microbial (and fine root) P uptake, but will increase the risk of soil P losses at the same time. Moreover, the expected warming effects on P pools and cycling processes were tested across soil depth and season to evaluate their consistency or their context dependence.

## Results and discussion
### Warming effects on soil P pools and P losses
After 14 years of warming, we found substantial decreases of total soil P (TP) from the 0–10 cm (−18%) and the 10–20 cm (−29%) soil layer in the warming treatment compared to the control treatment (Tables 1 and 2). There are different pathways potentially causing soil TP losses, i.e., plant P uptake, downward transportation of particulate and dissolved P, and P losses through soil erosion/runoff (Fig. 1).

In temperate forest ecosystems, elevated temperatures generally facilitate plant growth while warming globally had no effects on plant C:P ratios[4,23]. Greater biomass production indicates higher plant P uptake, which removes P from soil and allocates this to the plant compartment, in turn reducing the total soil P pools. Kwatcho Kengdo et al.[24], who studied fine root production, morphology, and element contents at the same site, reported a 128% increase in fine root production and a 17% increase in fine root biomass, but unaltered fine root P contents in the warming treatment compared to the control treatment. These results imply a higher P demand of the trees for fine root production, but they do not account for potentially greater plant P immobilization in aboveground tissues[17]. The potentially larger aboveground P stock should eventually return to the soils, which might (partially) compensate the soil P losses by fine root uptake and

aboveground P allocation in the warming treatment[25]. However, at Achenkirch site, due to the small plot size (2 by 2 meters) and soil-only warming (without canopy warming), the roots and canopies of trees are shared across the warming and the control treatments, and across experimental and non-experimental areas, causing comparable litter-fall P returns in warmed and un-warmed areas. Therefore, increased gross plant P uptake in the warming treatment combined with comparable plant litter P returns between treatments will result in the redistribution of P, ultimately reducing soil TP in warmed soils in the long run.

Soil P can also be lost from terrestrial systems as dissolved P and P that is bound to soil particles (e.g. colloid-associated P and particulate P) through subsurface flux[25,26]. High precipitation at this forest site (~1500 mm mean annual precipitation and ~800 mm seepage) could have favored the leaching of dissolved P[25,26], but this was not confirmed by soil solution chemistry (dissolved inorganic P in soil water was always below the detection limit at ~1 μM; data not shown). Colloidal particles (1–1000 nm size range) have a high specific surface area, which are typically composed of organic matter and metal oxyhydroxides, and thus can immobilize and transport a considerable amount of organic and inorganic P[27]. Moreover, colloid-associated P contributes up to 91% of total subsurface P fluxes in forest soils[27]. At our site, warming increased fine root production (+128%) more than fine root biomass (+17%), which implies a fast root turnover in warmed soil[24]. When roots die, their previous volume expansion can be preserved via the formation of macropores that might further generate effective and long-lasting preferential flow paths in soils, especially in undisturbed forests[25,26]. Preferential flow paths have been reported to significantly enhance colloidal and (particulate) dissolved P transport, due to fast transit times and therefore reduced interaction with the soil matrix[25–27]. Hereby, high precipitation at this forest site and the potential increase in root-induced preferential flow paths in warmed soils could have favored dissolved and colloid-associated P to migrate to deep soil layers, causing enhanced soil P losses in the warming treatment[25,27].

Moreover, subsurface P losses might be promoted by soil warming, as desorption processes increase stronger than sorption processes at higher temperatures[18–20]. Phosphate sorption shows little temperature sensitivity, with $Q_{10}$ values[28,29] between 1.0 and 1.1, while desorption reactions of inorganic and organic matter have $Q_{10}$ values of 1.2–2.0[18]. This may not only increase dissolved P losses, but also colloidal and particulate ones due to sorption of inorganic P ($P_i$) to colloidal and nano-particulate clays and metal oxyhydroxides, the latter of which significantly increased in warmed soils. Besides, substantial soil P losses can also occur by wind erosion and by water erosion[25,26], but erosional losses are likely not affected by warming in this experiment. Overall, (greater) net losses of TP from the warmed soils mainly resulted from increased net plant P uptake and subsurface dissolved and colloidal/particulate P fluxes.

### Warming effects on abiotic P processes
The potential temperature-induced increase in $P_i$ desorption relative to $P_i$ sorption was overprinted in situ as abiotic $P_i$ immobilization processes were significantly higher in warmed soils (Table 2), suggesting effective transfer of P from easily exchangeable pools through soil solution into strong geochemical sinks in the warming treatment. This formation of more recalcitrant P could have taken place via sorption (onto metal oxyhydroxides and clay minerals) and in calcareous soils via precipitation with calcium, forming Ca-apatite (Fig. 1)[8,9].

Soil metal oxyhydroxides are strong sorption sites for $P_i$. We observed higher amounts of dithionite-extractable iron (Fe) and crystalline Fe oxides in the warmed soils (Table 2). These higher Fe oxy-hydroxide contents in the warming treatment may be attributed to faster weathering processes[30], which result from elevated soil temperatures and increased fine root activity under warming[31].

**Table 1 | Comparison of soil properties at 0–10 cm and 10–20 cm depth in the warming and control treatments**

| Parameters | Units | Warming | | Control | |
|---|---|---|---|---|---|
| | | 0–10 cm | 10–20 cm | 0–10 cm | 10–20 cm |
| pH ($H_2O$) | | $6.59 \pm 0.09^a$ | $6.64 \pm 0.07^a$ | $7.03 \pm 0.06^b$ | $7.11 \pm 0.06^b$ |
| Soil water content | g $H_2O$ g d.s.$^{-1}$ | $0.80 \pm 0.03^b$ | $0.55 \pm 0.03^a$ | $0.98 \pm 0.05^c$ | $0.66 \pm 0.03^a$ |
| Soil organic C | g C kg d.s.$^{-1}$ | $115 \pm 5.82^c$ | $57.4 \pm 4.60^a$ | $123 \pm 5.96^c$ | $71.1 \pm 3.46^b$ |
| Total soil N | g N kg d.s.$^{-1}$ | $7.32 \pm 0.41^b$ | $4.13 \pm 0.35^a$ | $8.22 \pm 0.39^b$ | $4.89 \pm 0.24^a$ |
| Total soil P | g P kg d.s.$^{-1}$ | $0.65 \pm 0.04^b$ | $0.42 \pm 0.02^a$ | $0.79 \pm 0.04^c$ | $0.59 \pm 0.03^b$ |
| Microbial biomass C | mg C g d.s.$^{-1}$ | $2.26 \pm 0.17^{bc}$ | $1.21 \pm 0.11^a$ | $2.60 \pm 0.17^c$ | $1.85 \pm 0.24^b$ |
| Microbial biomass N | µg N g d.s.$^{-1}$ | $436 \pm 32.9^c$ | $214 \pm 25.5^a$ | $471 \pm 29.2^c$ | $315 \pm 39.5^b$ |
| Microbial biomass P | µg P g d.s.$^{-1}$ | $54.4 \pm 5.00^c$ | $22.4 \pm 3.02^a$ | $69.0 \pm 6.08^c$ | $35.9 \pm 4.66^b$ |
| Oxalate extracted Fe | mg FeO g d.s.$^{-1}$ | $0.84 \pm 0.10^a$ | $1.03 \pm 0.10^a$ | $0.95 \pm 0.11^a$ | $0.98 \pm 0.06^a$ |
| Oxalate extracted Mn | mg MnO g d.s.$^{-1}$ | $0.10 \pm 0.02^a$ | $0.07 \pm 0.01^a$ | $0.09 \pm 0.01^a$ | $0.06 \pm 0.01^a$ |
| Oxalate extracted Al | mg AlO g d.s.$^{-1}$ | $0.92 \pm 0.05^a$ | $0.83 \pm 0.05^a$ | $0.97 \pm 0.05^a$ | $0.82 \pm 0.06^a$ |
| Dithionite extracted Fe | mg Fe g d.s.$^{-1}$ | $3.99 \pm 0.40^a$ | $3.96 \pm 0.41^a$ | $3.54 \pm 0.15^a$ | $3.36 \pm 0.37^a$ |
| Dithionite extracted Mn | mg Mn g d.s.$^{-1}$ | $0.14 \pm 0.02^a$ | $0.12 \pm 0.02^a$ | $0.14 \pm 0.02^a$ | $0.11 \pm 0.01^a$ |
| Crystalline Fe oxides | mg Fe g d.s.$^{-1}$ | $3.15 \pm 0.37^a$ | $2.93 \pm 0.34^a$ | $2.59 \pm 0.13^a$ | $2.38 \pm 0.34^a$ |
| Sand content | % | $13.0 \pm 4.96^a$ | $16.8 \pm 5.06^a$ | $14.9 \pm 2.50^a$ | $24.3 \pm 5.59^a$ |
| Silt content | % | $54.8 \pm 3.25^a$ | $51.5 \pm 3.65^a$ | $54.1 \pm 2.83^a$ | $49.3 \pm 3.09^a$ |
| Clay content | % | $32.2 \pm 2.91^a$ | $31.8 \pm 2.37^a$ | $31.1 \pm 2.64^a$ | $26.4 \pm 3.06^a$ |
| Exchangeable $Na^+$ | mmol$_c$ kg d.s.$^{-1}$ | $0.72 \pm 0.07^{ab}$ | $0.52 \pm 0.08^a$ | $0.95 \pm 0.13^b$ | $0.46 \pm 0.09^a$ |
| Exchangeable $K^+$ | mmol$_c$ kg d.s.$^{-1}$ | $1.64 \pm 0.14^b$ | $0.85 \pm 0.10^a$ | $1.87 \pm 0.17^b$ | $0.89 \pm 0.10^a$ |
| Exchangeable $Ca^{2+}$ | mmol$_c$ kg d.s.$^{-1}$ | $405 \pm 31.5^{bc}$ | $262 \pm 33.8^a$ | $460 \pm 45.4^c$ | $278 \pm 31.7^b$ |
| Exchangeable $Mg^{2+}$ | mmol$_c$ kg d.s.$^{-1}$ | $175 \pm 14.8^b$ | $115 \pm 9.71^a$ | $189 \pm 18.6^b$ | $117 \pm 12.3^a$ |
| Exchangeable $Mn^{2+}$ | mmol$_c$ kg d.s.$^{-1}$ | $0.42 \pm 0.10^b$ | $0.08 \pm 0.02^a$ | $0.37 \pm 0.12^b$ | $0.05 \pm 0.00^a$ |
| Cation exchange capacity | mmol$_c$ kg d.s.$^{-1}$ | $583 \pm 45.8^{bc}$ | $379 \pm 43.0^a$ | $653 \pm 63.8^c$ | $397 \pm 43.7^b$ |
| Kaolinite | % | $60.5 \pm 5.28^a$ | $61.5 \pm 5.42^a$ | $61.2 \pm 5.35^a$ | $61.2 \pm 5.64^a$ |
| Illite | % | $11.0 \pm 1.53^a$ | $9.83 \pm 0.98^a$ | $10.3 \pm 1.31^a$ | $9.50 \pm 1.20^a$ |
| Chlorite and mixed layers | % | $28.5 \pm 6.42^a$ | $28.7 \pm 6.27^a$ | $28.5 \pm 6.10^a$ | $28.7 \pm 6.30^a$ |

Data presented are means±standard error, followed by different superscript letters (a, b, c, etc.), which indicate significant differences among treatments at two soil depths according to ANOVA followed by Tukey test. Soil organic C, microbial biomass C, N, and P, sand content, and exchangeable $Na^+$, $K^+$, and $Mn^{2+}$ were log-transformed to meet the assumptions prior to run the Tukey test. The sample size of pH, soil water content, soil organic C, total soil N, total soil P, and microbial biomass data is 18 (six blocks sampled in three seasons). The sample size of soil metal oxyhydroxide contents, soil texture, and exchangeable cations is 6 (six blocks). *d.s.* refers to dry soil.

Dithionite-extractable Fe was closely and positively correlated with abiotic P immobilization (Figs. 2 and 3a) and negatively correlated with (log-transformed) Olsen total P (Fig. 3b). Iron oxyhydroxides have often been reported to act as very strong sorption sites for $P_i$ in acidic soils, owing to their large specific surface area and positive surface charge, and this strong sorption capacity was reported to remain substantial at near neutral soil pH[11,32–34]. Moreover, the sorption capacity of Fe oxyhydroxides increases with decreasing soil pH[11,32,33]. Thus, the lower soil pH in the warming treatment can further have enhanced abiotic P immobilization (Table 2), thereby restricting $P_i$ availability in soil solution. Besides, the bonds between $P_i$ and Fe oxyhydroxides are highly inert, implying a slower P return via desorption to the soil solution and reducing soil bioavailable P within decadal warming[11,35].

Soil texture and clay mineralogy are also relevant for abiotic $P_i$ immobilization; especially clay minerals with a high specific surface area can form stable chemical bonds with P via calcium ($Ca^{2+}$) and magnesium ($Mg^{2+}$) bridges[10,36]. In this study, clay mineralogy did not change and was dominated by moderate P-sorbing kaolinites (~60%), chlorites associated with a mixed layered clay (~30%), and illites (~10%). Noteworthily, the sand content was significantly lower in warmed soil, whereas clay content was 11% higher than in control soil (though not significant). According to the results of PCA, abiotic $P_i$ immobilization was negatively associated with soil sand content, while it co-varied positively with soil clay content (Fig. 2). Soils with higher sand content commonly exert lower P sorption capacity as the dominant quartz particles in the sand size fraction have low surface area and a weak bonding affinity to P, whereas soils with high clay content often exhibit higher P sorption capacity, because of their large specific surface area and abundant bonding sites for P[36–38]. Thereby, the lower sand content in combination with a higher clay content in warmed soil can be another reason for intensified abiotic $P_i$ immobilization processes.

Short-term (decadal) warming effects on soil texture and mineralogy have rarely been studied. Elevated temperatures favor soil weathering processes[30,31], potentially increasing soil metal oxide contents at long time scales. Besides, higher temperatures can increase the soil clay contents[39,40]. However, these warming-induced changes are commonly observed at longer time scales, e.g. at centennial or millennial time periods[31,39,40]. Comparatively rapid (decadal) changes in soil metal oxides, soil texture and/or clay mineralogy have been shown in a glacier chronosequence study[41] and in other studies in response to changes in management and fertilizer regime[31,42–45]. If such rapid changes are triggered by soil warming as well, forest nutrient cycling can be negatively affected by increased sorptive constraints on substrate availability of these processes in the near future.

The forest soils studied here have high contents of exchangeable $Ca^{2+}$ and $Mg^{2+}$, a high cation-exchange capacity, and a base saturation close to 100% (Table 1). Considering the near-neutral soil pH (Table 1), $Ca^{2+}$ could play a major role in $P_i$ precipitation in the form of Ca-apatite, which is highly insoluble and therefore might act as another geochemical sink [$Ca_{10}(PO_4)_6(OH)_2$][46]. However, according to the X-ray diffractograms, the apatite content was very low at our site (Supplementary Fig. 1). Besides, long-term warming did not affect the exchangeable $Ca^{2+}$ pool (Table 2). Thus, the potential for

**Table 2 | Results of mixed-effects models with warming (W), soil depth (D), and season (S; for seasonal data) as fixed factors, block and warming duration as random factors, including interactive effects between main factors (seasonal data: *n* = 72; non-seasonal data: *n* = 24)**

| Parameter | Treatment | | | | Interaction (Sig. level) | | |
|---|---|---|---|---|---|---|---|
| | log response ratio | *F* value | *p* value | Sig. level | W x D | W x S | W x D x S |
| pH (H₂O) | −0.029 | 71.925 | <0.001 | *** | - | - | - |
| Soil water content | −0.085 | 25.039 | <0.001 | *** | - | - | - |
| Potential activity of acid phosphatase | 0.095 | 11.453 | 0.001 | ** | - | * | - |
| Total soil P | −0.113 | 37.556 | <0.001 | *** | - | - | - |
| Total soil inorganic P | −0.082 | 29.344 | <0.001 | *** | - | - | - |
| Total soil organic P | −0.112 | 34.439 | <0.001 | *** | - | - | - |
| Olsen P | 0.007 | 0.049 | 0.826 | - | - | * | - |
| Olsen inorganic P | −0.033 | 2.259 | 0.139 | - | - | - | - |
| Olsen organic P | 0.098 | 0.421 | 0.519 | - | - | - | - |
| Microbial biomass C | −0.108 | 15.168 | <0.001 | *** | - | - | - |
| Microbial biomass N | −0.082 | 12.355 | <0.001 | *** | * | - | - |
| Microbial biomass P | −0.135 | 28.472 | <0.001 | *** | - | - | - |
| Abiotic immobilization | 0.018 | 21.832 | <0.001 | *** | - | - | - |
| Total immobilization | 0.000 | 0.078 | 0.780 | - | - | - | - |
| Biotic immobilization | −0.075 | 32.820 | <0.001 | *** | - | - | - |
| Gross Pᵢ mobilization | −0.102 | 6.300 | 0.015 | * | - | - | - |
| Gross Pᵢ immobilization | −0.051 | 6.709 | 0.012 | * | - | - | - |
| Oxalate extracted Fe | −0.011 | 0.134 | 0.720 | - | - | | |
| Oxalate extracted Mn | 0.026 | 0.090 | 0.769 | - | - | | |
| Oxalate extracted Al | −0.010 | 0.201 | 0.659 | - | - | | |
| Dithionite extracted Fe | 0.062 | 4.999 | 0.041 | * | - | | |
| Dithionite extracted Mn | 0.011 | 0.095 | 0.762 | - | - | | |
| Crystalline Fe oxides | 0.088 | 4.665 | 0.047 | * | - | | |
| Sand | −0.120 | 5.533 | 0.033 | * | - | | |
| Silt | 0.012 | 0.721 | 0.409 | - | - | | |
| Clay | 0.047 | 1.651 | 0.218 | - | - | | |
| Exchangeable Na | −0.056 | 0.133 | 0.721 | - | - | | |
| Exchangeable K | −0.044 | 0.710 | 0.410 | - | - | | |
| Exchangeable Ca | −0.044 | 1.058 | 0.320 | - | - | | |
| Exchangeable Mg | −0.024 | 0.369 | 0.552 | - | - | | |
| Exchangeable Mn | 0.076 | 1.740 | 0.207 | - | - | | |
| Cation exchange capacity | −0.038 | 0.840 | 0.374 | - | - | | |
| Kaolinite | −0.004 | 0.555 | 0.468 | - | - | | |
| Illite | 0.021 | 1.572 | 0.229 | - | - | | |
| Chlorite and mix layers | 0.000 | 0.000 | 1.000 | - | - | | |

Significance levels are represented as asterisks: $^*p$ < .05; $^{**}p$ < .01; $^{***}p$ < .001; - not significant. Potential activity of acid phosphatase, total soil inorganic P, total dissolved P, dissolved inorganic P, dissolved organic P, microbial biomass C, microbial biomass N, microbial biomass P, biotic immobilization, gross Pᵢ immobilization, sand content, exchangeable Na, exchangeable K, and exchangeable Mn were log-transformed, and gross Pᵢ mobilization was sqrt-transformed before running mixed-effects model.

Ca²⁺-P precipitation does not play a large role in both, the warming and control treatments, and it does not explain the higher abiotic P immobilization in the warming treatment (Fig. 2).

Overall, increased abiotic P immobilization in warmed soils mainly resulted from increased P sorption onto Fe oxyhydroxides and clays. Phosphorus that is strongly bound to these minerals increases the risk of P losses through subsurface flow of colloids or fine particles, particularly via preferential flow[26,27]. The largest proportion of nano-colloids is formed from native Fe oxyhydroxides and clays in soils[47], and thereby may promote soil TP losses, instead of reducing them.

**Warming effects on biotic P processes**

Soil microbes compete with abiotic cycling processes for available P in soil solution (Fig. 1)[8,12]. Reduced soil total P after long-term warming likely intensified this competition between abiotic immobilization and microbial uptake in warmed soils. Moreover, enhanced abiotic Pᵢ immobilization in the warming treatment depleted available P for soil microbes. In a C-N-P substrate addition experiment, Shi et al.[48] measured the stimulation of microbial growth in response to substrate amendments as an indicator of microbial element limitation in the same soils in August 2019. They found a stronger stimulation of microbial growth in the combined C-P amendment than in the C-only addition in the warming treatment. In contrast, there was no difference in microbial growth stimulation between the combined C-P amendment and the C-only amendment in control soils. These results confirmed that, after 14 years of forest soil warming, soil microbes have increasingly become constrained by P in the warming treatment. Diminished P bioavailability, due to decreased TP and increased P sorption, remarkably reduced microbial biomass P (MBP; Tables 1 and 2)[49,50], which accounted for 10% of the total P losses. According to

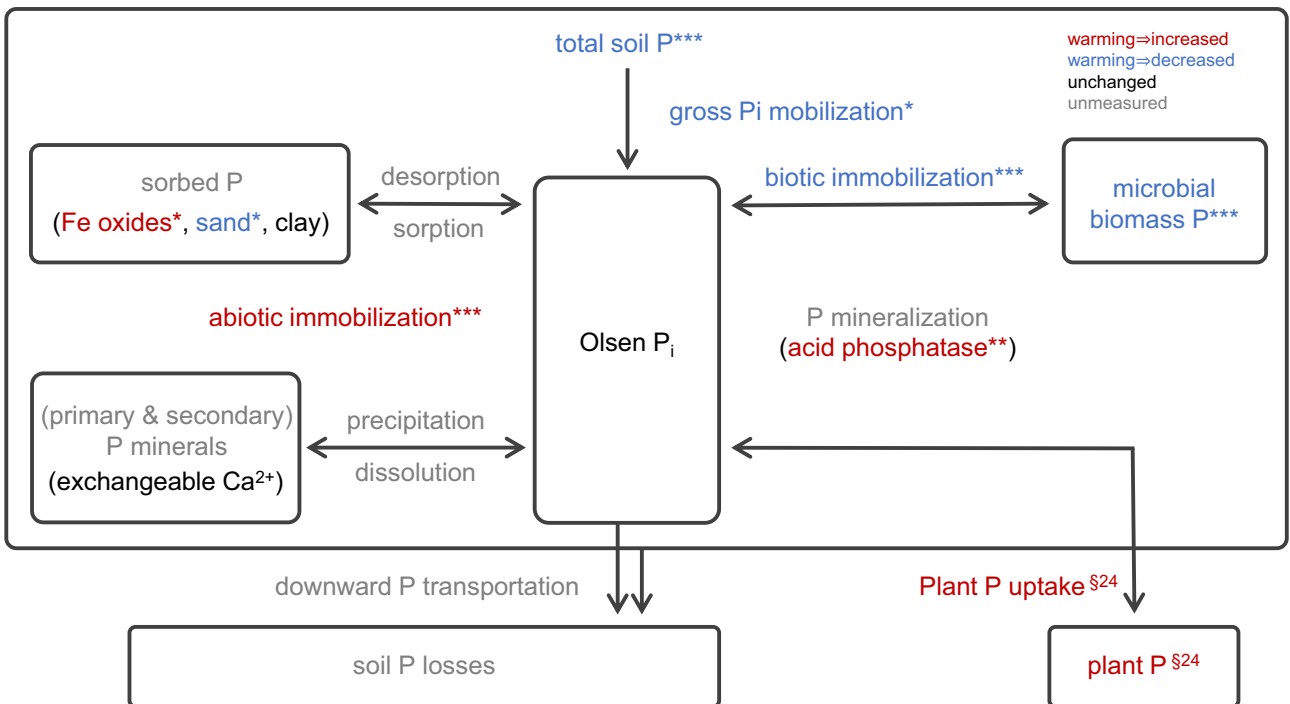

**Fig. 1 | Schematic illustration showing long-term warming effects on P pools, P processes, and geochemical parameters.** Rectangular boxes represent P pools and arrows represent P processes. Red, blue, and black texts indicate long-term warming increased, decreased, and unchanged parameters, respectively. Grey text indicates unmeasured parameters. Asterisks show the significance level of warming effects based on the results of mixed-effects models ($^*p < .05$; $^{**}p < .01$; $^{***}p < .001$). The results of plant P uptake are derived from Kengdo et al., 2022[24] from the same site in 2019.

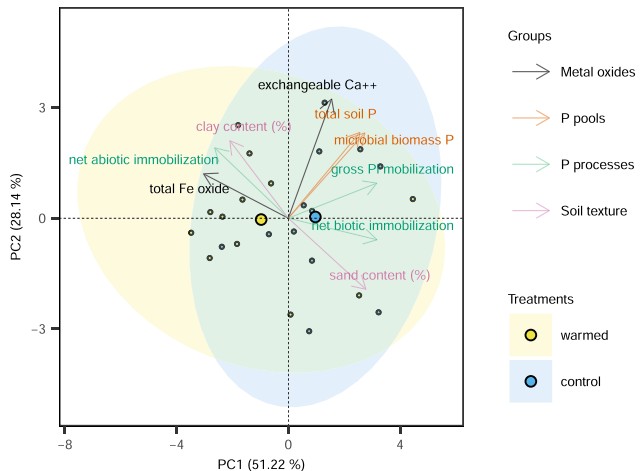

**Fig. 2 | Principal component analysis (PCA).** PCA was conducted with total soil P, microbial biomass P, exchangeable $Ca^{++}$, total Fe oxide, net abiotic immobilization, net biotic immobilization, gross $P_i$ mobilization, sand content (%), and clay content (%), with vectors displaying the association of these factors with principal component (PC) 1 and PC2. Individual cases are grouped by treatment, and large dots indicate the mean points of each group. Arrows represent loadings of variables. Positively correlated variables are grouped together, while negatively correlated variables are positioned on opposite sides of the plot origin. The distance between variables and the plot origin measures their importance on the respective PC.

the results of PCA and Pearson coefficient correlations, MBP was tightly and positively associated with total soil P, while it had weaker but significant negative relations with dithionite extractable Fe (Figs. 2 and 3c, d), implying stronger effects on MBP limitation from TP losses than from increased sorption (onto Fe oxyhydroxides).

In line with this, soil microbes also showed decreased net biotic immobilization rates (i.e., uptake) in the warming treatment (Table 2).

Restricted P availability in soil solution can stimulate soil microbes (and plants) to produce and excrete more extracellular phosphatases to acquire P that is bound in soil organic matter[8,10,11,51]. This accords with the significantly increased potential acid phosphatase activities in warmed soils (Table 2), which however could not compensate for the more constrained P availability as shown by lower MBP. Enzyme production is energy (i.e., C) and nutrient demanding for microbes[51]. Schindlbacher et al. reported[22] a ~40% higher soil $CO_2$ efflux in the warming treatment after 9 years of soil warming at the study site, which was still sustained when this study was performed in 2019 (unpublished data). Higher heterotrophic soil respiration implies less C (and nutrients) immobilized in microbial biomass, i.e., less growth investments and lower microbial C use efficiency, which is supported by the substantial decreases in microbial biomass C, N, and P in the warming treatment (Tables 1 and 2).

In conclusion, this study contributes novel data on warming effects on soil P processes, providing a more comprehensive understanding of climate change effects on the complex soil P cycle (Fig. 1). Our results suggest that long-term warming reduced bioavailable P, which resulted from substantial losses of soil TP (likely via plant P uptake and downward P transportation) and increased $P_i$ sorption and accumulation of P in soil recalcitrant fractions (onto Fe oxyhydroxides and clay minerals), can turn the soil microbes in temperate forest soils to become P limited. Moreover, according to the results of the mixed-effects models, most of the measured P pools and processes showed no interactions between warming and soil depth and/or season (Table 2), indicating consistent effects of long-term soil warming on the P cycle across different soil depths and seasons. In general, forest ecosystems on limestone and dolomite are P limited[52,53], due to the low P content of bedrock. Losses of P in these forests happen continuously by weathering, erosion and subsurface transport, and tree P demand is largely met via internal recycling, indicated by the upregulation of phosphatases and the shift from microbial P to plant fine root P pools under P limitation. The data therefore show that P constraints are likely

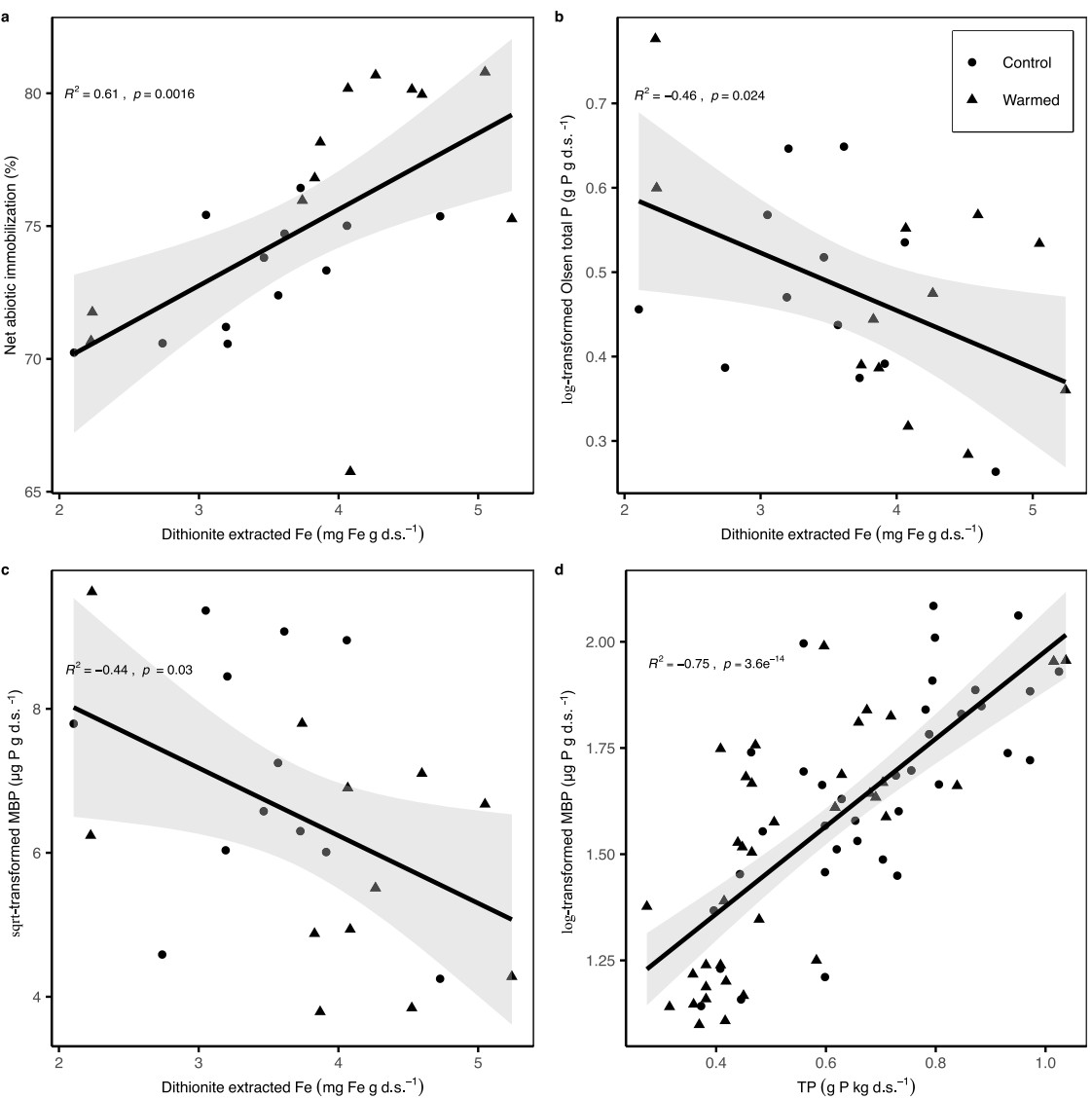

**Fig. 3 | Pearson correlations.** Correlations between **a** net abiotic P immobilization and dithionite extracted Fe ($n = 24$), **b** log-transformed Olsen total P and dithionite extracted Fe ($n = 24$), **c** sqrt-transformed microbial biomass P (MBP) and dithionite extracted Fe ($n = 24$), and **d** log-transformed MBP and total soil P (TP; $n = 72$). d.s. refers to dry soil.

amplified for both, microbes and plants, under warming. Given the strong signs that European forests have become increasingly P limited during the last decades[54,55], we therefore expect that the intensified competition between plants and soil microbes for this scarce resource will have negative consequences for plant net primary production and ecosystem C sequestration with climate warming. We however note that extrapolations of the soil warming effects found here to the ecosystem level needs to be done cautiously, given that we only warmed soils, and here only parts of a trees' root system, but not the aboveground compartment. Whole ecosystem warming might even further impair the delicate balance between P limited trees and soil microbes, given that warming increases the plant P demand.

## Methods
### Site description
The experimental site is located in a 140-year-old forest in the Northern Limestone Alps, Achenkirch, Austria (47°34´50″N, 11°38´21″E; 910 m a.s.l.). The forest is dominated by *Picea abies* (80%), *Fagus sylvatica* (15%), and *Abies alba* (5%), while the understory mainly consists of regeneration of *F. sylvatica*. The site has a cool and humid climate. In 2019, the mean monthly air temperature ranged from a high of

17.7 °C in June to a low of −4.5 °C in January, and annual precipitation was 1908 mm. Local mean annual air temperature and mean annual precipitation were 6.99 °C and 1493 mm (1988–2017), respectively (data from Zentralanstalt für Meteorologie und Geodynamik (ZAMG), Austria).

The bedrock is formed of dolomite. The soils are characterized as Chromic Cambisols and Rendzic Leptosols (based on the World Reference Base for Soil Resources[56]), with high carbonate content and near-neutral pH. The 0–10 cm soil increment largely represents the A horizon, and the depth increment of 10–20 cm is an AC horizon. The depth increment below 20–25 cm is constituted by the C horizon, with more than 80% represented by coarse dolomite rock. Moreover, soils exhibit a high spatial heterogeneity in texture (primarily determined by micro-topography) and resemble a small-scale mosaic of auto-chthonous young A/C profile soils. The soils contain older loamy moraine sediments, which were deposited by glaciers during the last ice age, as subsoils. Besides, in the clay fraction, our site was dominated by kaolinites (-60%), followed by chlorites associated with a mixed layered clay (-30%), and illites (-10%). The high contents of kaolinite in the clay fraction, a very stable clay mineral with moderate $P_i$ sorption capacity, clearly could not have formed during the last

deglaciation phase (10,000 to 15,000 years). That indicates inputs via aeolian deposition from older nearby soil formations and inputs from the Central Alps. Accordingly, soil texture showed high spatial variability within the different control and warming plots. The primary soil minerals are quartz, plagioclase, dolomite and K-feldspar at the study site (Supplementary Fig. 1). Moreover, goethite (iron hydroxide; $\alpha$-FeO(OH)) constitutes the main phase of crystalline Fe, ranging from 0.2 to 0.6 mass%, and amorphous iron (ferrihydrite) ranges from 0.07 to 0.19 mass%.

The experiment is an in-situ long-term soil warming manipulation experiment consisting of six blocks of paired control and warmed plots, randomly distributed at the site. In each block, warmed and control plots were established, each of $2 \times 2$ m size. Dependent on the treatment, resistance heating cables (0.4 cm diameter, TECUTE −0.18 Ohm-1 UV-1, Etherma, Austria) or dummy cables were buried at a depth of 3 cm, with a spacing of 7−8 cm. Temperature sensors (PT100; EMS, Brno, Czech Republic) were installed in the mineral soil at a depth of 5 cm between two heating cables in both warmed and control plots to enable a 4 °C increase in temperature of the warmed plots compared to their adjacent control plots. The heating system is only running during the snow-free season (April to November) to avoid substantial differences in environmental conditions between warming and control treatments due to warming-induced snow melting and adverse effects on soil hydrology. Long-term soil warming started by heating three paired plots in 2005, which was extended by adding further three paired plots in 2007.

### Soil sampling

Soils were sampled three times, in spring (2nd May), summer (6th August), and autumn (15th October) 2019 by stainless steel soil corers (diameter 2.5 cm). Six to seven cores per plot were taken randomly each time to obtain enough soil for later analyses. For the measurements of soil texture, metal oxyhydroxides, and clay mineralogy, another set of soil samples was collected in summer 2020 (18th August 2020). The soil cores were separated into 0−10 cm and 10−20 cm depth increments (Table 1), and the soils were then composited to form one mixed sample per depth, treatment, and block. All fresh soil samples were sieved (2 mm) and mixed homogenously. After this, warmed and control soil samples were incubated at the corresponding in-situ temperatures (warmed samples: 14 °C, 17 °C, and 14 °C; control samples: 10 °C, 13 °C, and 10 °C for spring, summer, and autumn, respectively) in incubators in the TER laboratories at the University of Vienna.

### Basic soil parameters

Soil pH was determined in a 1:5 (w:v) mix of air-dried soil and ultra-pure water using an ISFET pH sensor (Sentron, The Netherlands). Aliquots of 5 g fresh soil were dried at 105 °C (48 h) to gravimetrically measure soil water content.

Soil texture and particle-size distributions were determined using the combined wet sieving and sedimentation method following international guidelines for forest soil monitoring[57] and the Austrian standard ÖNORM L 1061-2 ("Soil physical investigations - Determination of particle size distribution of mineral soils in agriculture and forestry - Part 2: Fine soil"). Prior to particle-size fractionation and the determination of clay mineralogy, all samples were repeatedly treated with 15% $H_2O_2$ at 70 °C to remove soil organic matter, which is particularly high at this site.

For soil total N, we took two aliquots of oven-dried and ball-milled soil. One aliquot (-50 mg) was acidified repeatedly with 2 M HCl until all carbonates were removed. Soil organic C and N were then determined on both aliquots using an elemental analyzer coupled to an isotope ratio mass spectrometer (EA1110, CE Instruments and Delta V Advantage, Thermo Scientific). Exchangeable cations were analyzed after extraction with 0.1 M $BaCl_2$ solution using inductively coupled

plasma optical emission spectrometry (ICP-OES; Optima 8300, Perkin Elmer, Waltham, MA; OENORM 1086, 2014).

### P pools

Olsen total P (Olsen $P_t$), Olsen inorganic P (Olsen $P_i$), Olsen organic P (Olsen $P_o$), and MBP were determined using soil bicarbonate extraction and the malachite green method[58] with/-out previous acid persulfate digestion[59] and chloroform fumigation-extraction (CFE)[60]. Briefly, we weighed two aliquots of each soil sample (2 g fresh weight) into suitable sample containers. The first aliquots were fumigated with chloroform for 48 h, the second ones served as non-fumigated controls. Then, all soil samples were extracted with 0.5 M $NaHCO_3$ (pH = 8.5, 1:7.5 (w:v)) and filtered. All samples were acidified by addition of 2.75 M $H_2SO_4$ (10% of the extract volume), and subsamples of these extracts were digested by acid persulfate reagent (0.185 M sodium persulfate in 0.5 M $H_2SO_4$) with a ratio of extract: reagent of 20:3. Reactive inorganic $PO_4^{3-}$ of all samples was measured by the malachite green method[58]. Olsen $P_t$ and Olsen $P_i$ were calculated from the results of non-CFE digested and undigested aliquots, respectively. Olsen $P_o$ equals the difference between Olsen $P_t$ and Olsen $P_i$, while MBP was determined as the difference between digested CFE and digested non-CFE aliquots, applying a correction factor $K_{EP}$ of 0.4[61].

Soil total P (TP), total inorganic P (TIP), and total organic P (TOP) were determined by the ignition method[62]. Two aliquots (0.5 g) of finely ground soil samples were weighed. One aliquot was ignited at 450 °C for 5 h, oxidizing soil organic matter and thereby releasing organic P ($P_o$) as $P_i$ by high-temperature oxidation. Both aliquots, controls and ignited ones, were then extracted with 0.5 M $H_2SO_4$ (1:40 (w:v)) for 18 h and $PO_4^{3-}$ contents were measured after dilution with ultra-pure water with the malachite green method[58]. TP and TIP were then determined by the amount of $H_2SO_4$-extractable $P_i$ in the ignited and unignited soils, respectively, while TOP was calculated as the difference between TP and TIP.

### Microbial biomass

To determine microbial biomass C and N (MBC and MBN), we weighed two aliquots of fresh soil (2 g) in suitable sample containers. One aliquot was directly extracted with 1 M KCl (1:7.5 (w:v)), and dissolved organic C (DOC) and total dissolved N (TDN) were analyzed by a TOC/TN-Analyzer (TOC- VCPH/CPNT-NM-1, Shimadzu, Japan). Another aliquot was fumigated with chloroform for 48 h. The fumigated soil samples were also extracted and measured for DOC and TDN in the same way as for non-fumigated samples. MBC and MBN were calculated as the difference in DOC and TDN between the fumigated and the non-fumigated aliquots, using a correction factor ($K_{EC}$, $K_{EN}$) of 0.45[63,64].

### Soil oxyhydroxides

We weighed two aliquots of ground soil. One aliquot was mixed with 0.3 M $Na_3C_6H_5O_7 \cdot 2H_2O$ (w:v = 1:40) and 1 M $NaHCO_3$ (w:v = 1:10). Then, solid $Na_2S_2O_4$ (w:w = 1:1) was added, and the soil suspensions were shaken for 16 h (dithionite extractable Fe/Al/Mn). The extracts were centrifuged, filtered, and diluted 1:200 with ultra-pure water. Soil total Fe, Al and Mn were determined by ICP-OES (Optima 3200 xl, Perkin Elmer, Germany). The other aliquots were suspended in 0.2 M oxalate solution (pH 3: 0.1134 M $(NH_4)_2C_2O_4$ and 0.0866 M $C_2H_2O$; w-v = 1:50) and shaken for 2 h in the dark (oxalate extractable Fe/Al/Mn). The extracts were then filtered and diluted 1:500 with ultra-pure water. Soil amorphous (oxalate extractable) Fe, Al, and Mn were also measured by ICP-OES. Crystalline Fe oxides and oxyhydroxides were calculated by subtraction of amorphous Fe from total Fe.

### Clay mineralogy

The samples were studied by means of X-ray diffraction (XRD) using a Panalytical XPert Pro MPD diffractometer with automatic divergent slit, Cu LFF tube 45 kV, 40 mA, with an X´Celerator detector. The

measuring time was 100 s per step, with a stepsize of 0.017° 2 Theta. Dispersion of clay particles and destruction of organic matter was achieved by treatment with dilute hydrogen peroxide. Separation of clay fraction was carried out by using sieving and centrifugation methods. The exchange complex of each sample (<2 μm) was saturated with Mg and K using chloride solutions by shaking. The preferential orientation of the clay minerals was obtained by suction through a porous ceramic plate. Afterwards, expansion tests were made, using ethylenglycol, glycerol and DMSO as well as contraction tests heating the samples up to 550 °C. After each step, the samples were X-rayed from 2-40 °2θ. The clay minerals were identified according to Moore and Reynolds (1997)[65]. Semiquantitative estimations were carried out using the corrected intensities of characteristic X-ray peaks[66].

## Soil P cycling processes

Gross rates of $P_i$ mobilization and immobilization were analyzed using the $^{33}P$-isotope pool dilution approach as described by Wanek et al. (2019)[21]. Briefly, we prepared four aliquots of fresh soils (4 g). Two aliquots were sterilized twice (with 48 h in between) by autoclaving at 121 °C for 60 min. All aliquots, i.e., untreated fresh soils and sterilized soils (4 g), were then amended with 200 μL 20 kBq $^{33}P$ in the form of orthophosphoric acid. All samples were incubated at their corresponding field temperature. After 4 and 24 h, we stopped the incubations by extracting the samples with 20 mL 0.5 M $NaHCO_3$, followed by acidification of 3 mL extract with 0.3 mL 2.75 M $H_2SO_4$. Orthophosphate contents were measured by the malachite green method[58]. For liquid scintillation counting, the acidified samples were mixed with scintillation cocktail (Ultima Gold, Perkin Elmer; v:v = 1:16) and stored for 24 h in the dark. The $^{33}P$ activity was then measured by a Tri-Carb 1600 TR (Packard, Perkin Elmer, Austria). Gross rates of $P_i$ mobilization and immobilization were calculated based on Eqs. (1) and (2) for fresh soils.

$$\text{Gross Pi mobilization} = \frac{P_2 - P_1}{t_2 - t_1} \times \frac{\ln\left(\frac{SA_1}{SA_2}\right)}{\ln\left(\frac{P_2}{P_1}\right)} \tag{1}$$

$$\text{Gross Pi immobilization} = \frac{P_1 - P_2}{t_2 - t_1} \times \left[1 + \frac{\ln\left(\frac{SA_2}{SA_1}\right)}{\ln\left(\frac{P_2}{P_1}\right)}\right] \tag{2}$$

where $t_1$ and $t_2$ are the termination time points 1 and 2 (4 and 24 h), $P_1$ and $P_2$ are the corresponding contents (μg P g$^{-1}$ d.s.) of soil inorganic P at $t_1$ and $t_2$, and $SA_1$ and $SA_2$ are the specific activities (in Bq μg$^{-1}$ P) at the two time points.

We estimated net total, net abiotic, and net biotic immobilization based on the recovery of the added tracer in fresh and sterilized soil aliquots. Net total $P_i$ immobilization equals 100% of $^{33}P_i$ added minus the percentage of $^{33}P$ recovered in live soils. Net abiotic immobilization equals 100% of $^{33}P_i$ added minus the percentage of $^{33}P$ recovered in sterilized soils. Biotic net immobilization was estimated as the difference between total net immobilization and abiotic net immobilization.

The potential activity of acid phosphatase (AP) was determined fluorimetrically with the substrate addition method[67]. Briefly, soil slurries were prepared by mixing and ultrasonicating fresh soil with 50 mM sodium acetate buffer (pH=5.5, 1:100 (w:v), energy input = 350 J ± 20 J). Then, the soil slurries were transferred to black microtiter plates, each sample occupying one column (i.e., eight slots per sample). To the first three aliquots of each soil slurry enzyme substrate, i.e., 4-methylumbelliferyl-phosphate, was added. The other five aliquots were spiked with different concentrations of methylumbelliferone (MUF) to correct for fluorescence quenching in each soil sample. All microtiter plates were then incubated at their corresponding field temperature in the dark for 30–180 min, with repeated fluorescence

measurements every 30 min. The fluorimetric measurements were done using a TECAN Infinite® M200 spectrophotometer, with an excitation and emission wavelengths of 365 nm and 450 nm, respectively. The potential enzyme activities were estimated based on the concentrations of released MUF during the measurement period and calibrated by sample-specific standards.

## Data analyses

All statistical analyses were performed using R 3.6.3. Data were log- or sqrt-transformed before analyses to meet the assumptions of normality and homoscedasticity, if necessary. We evaluated the homogeneity of variance using the Bartlett test, and normal distribution by checking histograms and boxplots visually, and finally Q-Q plots of the residuals of linear mixed-effects models. We applied linear mixed-effects models (lme4 package) with warming, soil depth, and season as fixed factors, and block and warming duration as random factors, to test their individual effects and interactions on all measured parameters. Moreover, we performed multiple comparisons within a factor with Tukey HSD post hoc tests using the multcomp package. Regarding the data of soil texture, metal oxyhydroxides, and clay mineralogy, we set warming and soil depth as fixed factors and block and duration as random factors. Pearson correlations were run to investigate possible univariate relationships in more depth. For the correlations between the data from the three seasons in 2019 and the soil data from Aug 2020, we averaged the seasonal data and then ran Pearson correlations against the data from 2020. The significance threshold was set to 0.05 for all the analyses mentioned above. Moreover, principal component analysis (PCA) was implemented to visualize the potential relationships between those parameters, which were strongly affected by warming, but were not closely co-varying, using the FactoMineR and factoextra packages in R.

## Data availability

The datasets generated and analyzed in this study are available in the DRYAD repository: https://doi.org/10.5061/dryad.9p8cz8wk8.

## Code availability

The R codes used to complete the analysis and generate figures in this study are available in the following online repository: https://doi.org/10.5281/zenodo.7536822.

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

## Acknowledgements

This study was funded by the Austrian Science Fund (FWF; project I 3745). We sincerely thank Christian Holtermann for field site maintenance, Margarete Watzka, Sabine Maringer, Sabrina Pober, Ludwig Seidl, and Renate Krauß for technical and material support, and Johann Püspök for laboratory guidance. Moreover, we acknowledge the inspirational communications and mutual help from people in the TER (Terrestrial Ecosystem Research) laboratories at the University of Vienna.

## Author contributions

W.W, A.S, W.B, and E.I. designed the study. Y.T., C.S., C.U.M., S.K.K., J.H., E.I., W.B., A.S., and W.W. implemented the field experiment. Y.T., C.S., C.U.M., S.K.K., F.O., K.M., and W.W. conducted the lab experiment. Y.T. performed data analysis and wrote the manuscript, with the contributions of all co-authors.

## Competing interests

The authors declare no competing interests.
