## [Peer Review File · Nature Communications]

Long-term soil warming decreases microbial phosphorus utilization by increasing abiotic phosphorus sorption and phosphorus lossesREVIEWER COMMENTS

Reviewer #1 (Remarks to the Author):

The authors use a 14-year soil warming experiment in a mountain forest in Austria, to investigate soil P pools and soil P cycling in response to warming. The authors demonstrate decreased P mobilisation and increased P immobilisation, despite an increase in activity of phosphatase enzymes. The authors then present a framework of mechanisms on how P immobilisation is correlated to the abundance of Fe-Al oxyhydroxides; and how in turn this may be related to root production and microbial P dynamics (biomass immobilisation and phosphatase activity; although the latter two are from different manuscripts published elsewhere and only summarised here).

I found the results very interesting and think they will provide a valuable contribution to the literature. While the experiment and data appear robust, I think a substantial revision is needed for the paper to be worthy of publication. In its current format the manuscript fails to provide sufficient background to provide a rationale for the presented hypotheses, or to place the results in the context of the wider literature. In places the paper reads more like a summary/synthesis of this specific experiment (drawing on other papers currently in review elsewhere) rather than reporting a new finding in the context of the wider literature.

For example, the final conclusions paragraph is a summary of the results from this experiment and no effort is made to put the results in the context of the literature, or to present the wider relevance and context of the findings - the paragraph includes no citations of other work.

I am also concerned about how the results here are framed as representing a comparison of 'topsoil' vs. 'subsoil', based on a very marginal depth difference of 0-10cm vs 10-20cm. Similarly, for the 'seasonal' patterns there was no sampling during winter.

The main findings reported (Table 1, Figure 1-2) are indeed interesting. However, it must be emphasised that all the reported results are based on correlations only. Therefore, greater caution is needed to describe these as correlations rather than causal relationships when inferring mechanisms in the discussion.

Specific comments:

Title: I'm not sure 'impairs' is an appropriate description of the results. Impairs means to diminish in function, which I don't think is reflected here in terms of P cycling. The manuscript shows interesting changes in P cycling, with immobilisation and increased enzyme investment, but I don't think this can be summarised as being an impaired.

Line 34: 'By the end' instead of 'Until'

Lines 48-49: Sentence doesn't make sense to me - Please check language here ('did either promote or have no effects').

Lines 51-53: Would be improved here by describing in more detail the studies have looked at P dynamics in warmed soils, and being more explicit about what is novel here. For example, recent paper on P mobilisation in warmed soils by Zhou et al: <https://doi.org/10.1111/gcb.15914>.

Lines 59-60: The rationale behind studying changes across soil depth is very insubstantial. If soil depth is an important part of this study then more detail is needed on why responses in deep soils by differ, providing rationale for hypotheses on response differences across soil depths.

Lines 60-61: As above for soil depth, scarce information here on why seasonality is important in the context of this study, to provide a rationale for any hypotheses on seasonal dynamics. Suggest specifying what properties change seasonally and explicitly framing hypotheses on those seasonal

changes (e.g. temperature/moisture – how change seasonally, how will this affect P dynamics under warming?).

Lines 63-64: Related to the above here it is stated that the overall objective of this study is to investigate how warming affects soil P... 'across soil depths and seasons'. Given this overall objective, please include more detail in the introduction on why depth/season is important (both for this study site – and in the context of other study sites).

Line 74: The soil depths for this study are 0-10cm and 10-20cm. What soil horizons do these represent? I would have thought they are both representative of O or O-A horizons. Please include this information for these soils. Are soil properties across this 10cm depth differences sufficiently different to warrant using this as an experimental comparison? If there is a large difference, then it should be explained here so the reader understands that 0-10 cm vs. 10-20cm represents a valid comparison of different soil environments.

Line 74: The seasonal comparison here is May, August, October; and includes no observation during the winter. I think some justification is needed to explain why these three months were selected, and when discussing seasonal dynamics it should be mentioned that no observations were made during the coldest months.

Line 80: Related to above comment, here the '10-20cm' treatment is described as 'subsoil'. I would suggest subsoil would come from the B horizon (usually much deeper, at least >30cm), and have significantly less organic matter than the top soil. Please present properties on these two soil horizon treatments, to provide a justification for their use as a treatment comparison.

Line 83: Please describe how and why warming 'facilitates plant P uptake from soil to reduce soil P pools'?

Lines 87-91: Suggest review and re-write this text with greater focus. In its present form I find this text confusing and appears as a straw-man argument. Initially it is suggested that the results imply higher P demand and greater P uptake; but in the following sentence it is stated that root P contents did not change. Therefore, if root P contents did not change, isn't this evidence against greater P uptake?

Lines 92-106: A common problem in this manuscript is the lack of evidence drawn from the literature, to place the reported results in a wider context. This paragraph is one example of that. Many conjectures are made (e.g. lines 100-103) with no supporting literature cited.

Lines 123-127: Here and elsewhere, greater caution is needed when inferring mechanisms based on correlations. (e.g. here a direct mechanism is inferred based on a correlation 'increased Fe oxyhydroxide contents in warmed soils contributed to a greater abiotic P immobilisation strength').

Lines 140-150: This text about the geology of the study sites appears out of place here, perhaps this would be better placed in the methods or site description.

Lines 150-160: This text needs a significant re-write to remove speculation and to base the discussion on the results and grounded in the wider literature. At present the text includes no references to other studies, but is full of speculation without any support.

e.g. 'the trend towards'.. 'might have contributed, but we cannot attributed this to a warming effect'. 'It seems rather unlikely that warming affected soil texture, but it may not be totally ruled out...' It's fine to have some speculation, but it needs to be firmly grounded in a hypothesis based on evidence (if not evidence from this study, then evidence from elsewhere).

Lines 177-191: This paragraph appears to summarise results from another study currently in review elsewhere.

Overall, as I read this paper I'm confused as to whether it's a results paper or a review/summary paper (either are fine – but I would suggest the authors consider which to frame this as during

revision).

Lines 205-222: The conclusions paragraph is another example of the lack of depth here. The paragraph reads as a list of results and includes no references to other work.

Reviewer #2 (Remarks to the Author):

GENERAL COMMENTS:

This paper addresses an important topic in soil science: contributes to a better understanding of soil P biotic and abiotic cycling under main drivers of global change. In particular the work tries to shed some light on the effect of warming over abiotic processes (weathering, P sorption), processes that are very often overlooked on soil warming experiments (commonly centered in understanding biological responses).

A noteworthy result of this manuscript is the relation between the decrease of total soil P pools under warming with an increased sorption strength for inorganic P to iron oxyhydroxides and clay. This work proves that abiotic processes are important as biotic ones as drivers of important changes in nutrient cycling under global change. This work is of significance to the field and related fields. The processes described are not new (soil P pathways organic and inorganic cycling). However, the novelty of this work consists in applying a very complete set of measurements to a mid-term (14 years) warming experiment providing a complete in-depth study of soil P cycle.

This manuscript addresses topics of main-line interest to Nature Communications, it is well-written and thoroughly referenced. The title is appropriate, the abstract conveys the main points, and the overall conclusions are of wide interest. However, I have concerns on some points of the discussion and I miss a better graphical (Figures) support. If the points addressed bellow are improved enough I would recommend its publication by the requested journal.

I would ask the authors why they refer to dissolved inorganic/organic P instead of the commonly used "available P" or P-Olsen. DOP and DIP terms result confusing as they are mainly terms applied to water studies.

The authors affirm on page 4 (line 69) that inorganic P sorption decreases at higher temperatures and include a reference that support this statement. However, most literature demonstrate that sorption processes are higher at higher temperatures (when these are close to ambient temperatures 5-25°C).

From my point of view, it is missing in the text a better description of the soil type (classification, mineralogy, horizons depth) on the main text to provide elements for a more comprehensive discussion. At certain part of the discussion the authors refer to a high spatial heterogeneity in texture of the sampled soils and the presence of loamy moraine sediments as subsoils. This information would provide a better understanding of the site and soil processes if introduced before together with more accurate descriptions.

The authors argue possible biotic pathways to explain the decrease of total P pools. This includes a higher uptake by plants explained by an increase in fine root production. What are the implications and limitations of the experimental design to evaluate plant response to warming? In contrast to what happens with current climate change, in this case the soil is warmed but not the aboveground which restricts or unbalances plant response to belowground adaptations.

Higher Fe hydroxide content after warming treatments is attributed to faster weathering processes. This affirmation is surprising as a result of a 14-years warming. This is because weathering intensity directly relates to temperature, CO₂ concentration and precipitation but it is generally attributed to longer time scales (millennial). There aren't many published studies considering the role of weathering changes at decadal time scales. I would appreciate that the authors develop more this

idea including more references. At the moment the effects of the climate-change induced changes in weathering are not well studied and understood.

Apart from clay mineralogy, what are the primary minerals? Which is the mineral phase attributed to the crystalline Iron? When compared with the total mineralogy (not only clays), is it possible to estimate the % of amorphous and crystalline Iron?

Figures can be improved. Figure 1 can incorporate the complete variable name on PCA loadings. It is recommendable to use color blind friendly palettes (easy available in R). More elements for discussion could be derived from Figure 1, as the meanings or interpretation for PC1 or PC2. On Figure 2, samples could be distinguished among treatments. I recommend the authors to include a third and final Figure with a schematic representation of the main processes described. I would consider this third Figure essential to achieve publication in the requested journal.

I encourage the authors to reinforce in a clearer way what is new from their contribution, and to remark its importance in relation to the global understanding of soil carbon and nutrient cycling.

SPECIFIC COMMENTS

Pag 3 line 35: current warming is already above 1°C above pre-industrial levels

Pag 5 line 100: Soil water loss is expected to be higher at the top soil than at the subsoil, is it like this? Can this relate to changes on DOP and DIP between different depths through a higher reduction of diffusion capacity on the top soil?

Pag 7 line 139: First time mention to elements of soil description, better mention them earlier.

Pag 7 line 144: better “moderate” instead of high Pi sorption capacity when referring to kaolinite. It has lower sorption capacity than 2:1 clay. The same for line 175 when is mentioned “strong P-sorbing” kaolinite.

Pag 8 line 165: In relation with a previous comment asking for more information on mineralogy, are there proves of apatite precipitation on a general diffractogram?

Pag, first paragraph: Could you discuss or compare the effect of inorganic P sorption vs. the increased root uptake when considering P limitation for the microbial pool? (a limitation that in turn may explain the microbial pool decrease under warming).

Methodology: Please specify why the sampling design was performed at 0-10 and 10-20 and how relates with soil horizons and composition.

Please specify protocol or references for chloroform fumigation-extraction applied (line 287).

REVIEWER COMMENTS

Reviewer #1 (Remarks to the Author):

The authors use a 14-year soil warming experiment in a mountain forest in Austria, to investigate soil P pools and soil P cycling in response to warming. The authors demonstrate decreased P mobilisation and increased P immobilisation, despite an increase in activity of phosphatase enzymes. The authors then present a framework of mechanisms on how P immobilisation is correlated to the abundance of Fe-Al oxyhydroxides; and how in turn this may be related to root production and microbial P dynamics (biomass immobilisation and phosphatase activity; although the latter two are from different manuscripts published elsewhere and only summarised here).

I found the results very interesting and think they will provide a valuable contribution to the literature. While the experiment and data appear robust, I think a substantial revision is needed for the paper to be worthy of publication. In its current format the manuscript fails to provide sufficient background to provide a rationale for the presented hypotheses, or to place the results in the context of the wider literature. In places the paper reads more like a summary/synthesis of this specific experiment (drawing on other papers currently in review elsewhere) rather than reporting a new finding in the context of the wider literature.

For example, the final conclusions paragraph is a summary of the results from this experiment and no effort is made to put the results in the context of the literature, or to present the wider relevance and context of the findings - the paragraph includes no citations of other work.

I am also concerned about how the results here are framed as representing a comparison of 'topsoil' vs. 'subsoil', based on a very marginal depth difference of 0-10cm vs 10-20cm. Similarly, for the 'seasonal' patterns there was no sampling during winter.

The main findings reported (Table 1, Figure 1-2) are indeed interesting. However, it must be emphasised that all the reported results are based on correlations only. Therefore, greater caution is needed to describe these as correlations rather than causal relationships when inferring mechanisms in the discussion.

Specific comments:

Title: I'm not sure 'impairs' is an appropriate description of the results. Impairs means to diminish in function, which I don't think is reflected here in terms of P cycling. The manuscript shows interesting changes in P cycling, with immobilisation and increased enzyme investment, but I don't think this can be summarised as being an impaired.

Line 34: 'By the end' instead of 'Until'

Lines 48-49: Sentence doesn't make sense to me - Please check language here ('did either promote or have no effects').

Lines 51-53: Would be improved here by describing in more detail the studies have looked at P dynamics in warmed soils, and being more explicit about what is novel here. For example, recent paper on P mobilisation in warmed soils by Zhou et al: <https://doi.org/10.1111/gcb.15914>.

Lines 59-60: The rationale behind studying changes across soil depth is very insubstantial. If soil depth is an important part of this study then more detail is needed on why responses in deep soils differ, providing rationale for hypotheses on response differences across soil depths.

Lines 60-61: As above for soil depth, scarce information here on why seasonality is important in the context of this study, to provide a rationale for any hypotheses on seasonal dynamics. Suggest specifying what properties change seasonally and explicitly framing hypotheses on those seasonal changes (e.g. temperature/moisture – how change seasonally, how will this affect P dynamics under warming?).

Lines 63-64: Related to the above here it is stated that the overall objective of this study is to investigate how warming affects soil P... 'across soil depths and seasons'. Given this overall objective, please include more detail in the introduction on why depth/season is important (both for this study site – and in the context of other study sites).

Line 74: The soil depths for this study are 0-10cm and 10-20cm. What soil horizons do these represent? I would have thought they are both representative of O or O-A horizons. Please include this information for these soils. Are soil properties across this 10cm depth differences sufficiently different to warrant using this as an experimental comparison? If there is a large difference, then it should be explained here so the reader understands that 0-10 cm vs. 10-20cm represents a valid comparison of different soil environments.

Line 74: The seasonal comparison here is May, August, October; and includes no observation during the winter. I think some justification is needed to explain why these three months were selected, and when discussing seasonal dynamics it should be mentioned that no observations were made during the coldest months.

Line 80: Related to above comment, here the '10-20cm' treatment is described as 'subsoil'. I would suggest subsoil would come from the B horizon (usually much deeper, at least >30cm), and have significantly less organic matter than the top soil. Please present properties on these two soil horizon treatments, to provide a justification for their use as a treatment comparison.

Line 83: Please describe how and why warming 'facilitates plant P uptake from soil to reduce soil P pools'?

Lines 87-91: Suggest review and re-write this text with greater focus. In its present form I find this text confusing and appears as a straw-man argument. Initially it is suggested that the results imply higher P demand and greater P uptake; but in the following sentence it is stated that root P contents did not change. Therefore, if root P contents did not change, isn't this evidence against greater P uptake?

Lines 92-106: A common problem in this manuscript is the lack of evidence drawn from the literature, to place the reported results in a wider context. This paragraph is one example of that. Many conjectures are made (e.g. lines 100-103) with no supporting literature cited.

Lines 123-127: Here and elsewhere, greater caution is needed when inferring mechanisms based on correlations. (e.g. here a direct mechanism is inferred based on a correlation 'increased Fe oxyhydroxide contents in warmed soils contributed to a greater abiotic P immobilisation strength').

Lines 140-150: This text about the geology of the study sites appears out of place here, perhaps this could be better placed in the methods or site description.

Lines 150-160: This text needs a significant re-write to remove speculation and to base the discussion on the results and grounded in the wider literature. At present the text includes no references to other studies, but is full of speculation without any support. e.g. 'the trend towards'.. 'might have contributed, but we cannot attributed this to a warming effect'. 'It seems rather unlikely that warming affected soil texture, but it may not be totally ruled out...'

It's fine to have some speculation, but it needs to be firmly grounded in a hypothesis based on evidence if not evidence from this study, then evidence from elsewhere).

Lines 177-191: This paragraph appears to summarise results from another study currently in review elsewhere. Overall, as I read this paper I'm confused as to whether it's a results paper or a review/summary paper (either are fine – but I would suggest the authors consider which to frame this as during revision).

Lines 205-222: The conclusions paragraph is another example of the lack of depth here. The paragraph reads as a list of results and includes no references to other work.

Reviewer #2 (Remarks to the Author):

GENERAL COMMENTS:

This paper addresses an important topic in soil science: contributes to a better understanding of soil P biotic and abiotic cycling under main drivers of global change. In particular the work tries to shed some light on the effect of warming over abiotic processes (weathering, P sorption), processes that are very often overlooked on soil warming experiments (commonly centered in understanding biological responses).

A noteworthy result of this manuscript is the relation between the decrease of total soil P pools under warming with an increased sorption strength for inorganic P to iron oxyhydroxides and clay. This work proves that abiotic processes are important as biotic ones as drivers of important changes in nutrient cycling under global change. This work is of significance to the field and related fields. The processes described are not new (soil P pathways organic and inorganic cycling). However, the novelty of this work consists in applying a very complete set of measurements to a mid-term (14 years) warming experiment providing a complete in-depth study of soil P cycle.

This manuscript addresses topics of main-line interest to Nature Communications, it is well-written and thoroughly referenced. The title is appropriate, the abstract conveys the main points, and the overall conclusions are of wide interest. However, I have concerns on some points of the discussion and I miss a better graphical (Figures) support. If the points addressed bellow are improved enough I would recommend its publication by the requested journal.

I would ask the authors why they refer to dissolved inorganic/organic P instead of the commonly used "available P" or P-Olsen. DOP and DIP terms result confusing as they are mainly terms applied to water studies.

The authors affirm on page 4 (line 69) that inorganic P sorption decreases at higher temperatures and include a reference that support this statement. However, most literature demonstrate that sorption processes are higher at higher temperatures (when these are close to ambient temperatures 5-25°C).

From my point of view, it is missing in the text a better description of the soil type (classification, mineralogy, horizons depth) on the main text to provide elements for a more comprehensive discussion. At certain part of the discussion the authors refer to a high spatial heterogeneity in texture of the sampled soils and the presence of loamy moraine sediments as subsoils. This information would provide a better understanding of the site and soil processes if introduced before together with more accurate descriptions.

The authors argue possible biotic pathways to explain the decrease of total P pools. This includes a higher uptake by plants explained by an increase in fine root production. What are the implications and limitations of the experimental design to evaluate plant response to warming? In contrast to what happens with current climate change, in this case the soil is warmed but not the aboveground which restricts or unbalances plant response to belowground adaptations.

Higher Fe hydroxide content after warming treatments is attributed to faster weathering processes. This affirmation is surprising as a result of a 14-years warming. This is because weathering intensity directly relates to temperature, CO₂ concentration and precipitation but it is generally attributed to longer time scales (millennial). There aren't many published studies considering the role of weathering changes at decadal time scales. I would appreciate that the authors develop more this idea including more references. At the moment the effects of the climate-change induced changes in weathering are not well studied and understood.

Apart from clay mineralogy, what are the primary minerals? Which is the mineral phase attributed to the crystalline Iron? When compared with the total mineralogy (not only clays), is it possible to estimate the % of amorphous and crystalline Iron?

Figures can be improved. Figure 1 can incorporate the complete variable name on PCA loadings. It is recommendable to use color blind friendly palettes (easy available in R). More elements for discussion could be derived from Figure 1, as the meanings or interpretation for PC1 or PC2.

On Figure 2, samples could be distinguished among treatments. I recommend the authors to include a third and final Figure with a schematic representation of the main processes described. I would consider this third Figure essential to achieve publication in the requested journal.

I encourage the authors to reinforce in a clearer way what is new from their contribution, and to remark its importance in relation to the global understanding of soil carbon and nutrient cycling.

SPECIFIC COMMENTS

Pag 3 line 35: current warming is already above 1°C above pre-industrial levels

Pag 5 line 100: Soil water loss is expected to be higher at the top soil than at the subsoil, is it like this? Can this relate to changes on DOP and DIP between different depths through a higher reduction of diffusion capacity on the top soil?

Pag 7 line 139: First time mention to elements of soil description, better mention them earlier.

Pag 7 line 144: better "moderate" instead of high Pi sorption capacity when referring to kaolinite. It has lower sorption capacity than 2:1 clay. The same for line 175 when is mentioned "strong P-sorbing" kaolinite.

Pag 8 line 165: In relation with a previous comment asking for more information on mineralogy, are there proves of apatite precipitation on a general diffractogram?

Pag, first paragraph: Could you discuss or compare the effect of inorganic P sorption vs. the increased root uptake when considering P limitation for the microbial pool? (a limitation that in turn may explain the microbial pool decrease under warming).

Methodology: Please specify why the sampling design was performed at 0-10 and 10-20 and how relates with soil horizons and composition.

Please specify protocol or references for chloroform fumigation-extraction applied (line 287).

We are grateful to the reviewers' comments, which helped us improve the manuscript. We addressed most comment individually and several highly connected comments together. The responses are in green for clarity, and corresponding changes in the manuscript are highlighted in yellow in the revised manuscript. Besides, we carefully followed the guide to formatting articles for Nature Communications, and made minor changes (highlighted in orange) in the revised manuscript and in the revised display files.

REVIEWER COMMENTS

Reviewer #1 (Remarks to the Author):

The authors use a 14-year soil warming experiment in a mountain forest in Austria, to investigate soil P pools and soil P cycling in response to warming. The authors demonstrate decreased P mobilisation and increased P immobilisation, despite an increase in activity of phosphatase enzymes. The authors then present a framework of mechanisms on how P immobilisation is correlated to the abundance of Fe-Al oxyhydroxides; and how in turn this may be related to root production and microbial P dynamics (biomass immobilisation and phosphatase activity; although the latter two are from different manuscripts published elsewhere and only summarised here).

I found the results very interesting and think they will provide a valuable contribution to the literature. While the experiment and data appear robust, I think a substantial revision is needed for the paper to be worthy of publication. In its current format the manuscript fails to provide sufficient background to provide a rationale for the presented hypotheses, or to place the results in the context of the wider literature. In places the paper reads more like a summary/synthesis of this specific experiment (drawing on other papers currently in review elsewhere) rather than reporting a new finding in the context of the wider literature.

For example, the final conclusions paragraph is a summary of the results from this experiment and no effort is made to put the results in the context of the literature, or to present the wider relevance and context of the findings - the paragraph includes no citations of other work.

1-1. Author response:

We thank the reviewer for the positive comments and feedback. We integrated all the comments and suggestions in the revised manuscript. For the detailed responses, please see below.

I am also concerned about how the results here are framed as representing a comparison of 'topsoil' vs. 'subsoil', based on a very marginal depth difference of 0-10cm vs 10-20cm. Similarly, for the 'seasonal' patterns there was no sampling during winter.

1-2. Author response:

We added a new table (Table 1, see the revised Display document) showing the comparison of soil properties from 0-10 cm and 10-20 cm soil depth in the warming and control treatments. According to this table, the readers can see the substantial difference between these two soil layers, e.g. the soil organic carbon content is almost double at 0-10 cm compared to 10-20 cm soil depth. Besides, we replaced "top- and subsoil" with "0-10 and 10-20 cm" to be more accurate. Moreover, these soils are

very shallow, being Chromic Cambisols and Rendzic Leptosols, where at most spots in this forest one arrives at the C horizon at only 20-25 cm depth.

Regarding the exclusion of winter sampling, the reason is that we only heat up the warming plots during the snow-free seasons, i.e. during winter time warmed plots are at ambient temperature. There is snow cover (up to 1 m) at our site during winter months. Snow cover can act as thermal insulation layer, which keeps the soil temperatures of control plots always above 0 °C. As the soil temperatures in warming plots are based on the soil temperatures of their adjacent control plots plus 4 °C, these higher soil temperatures would melt the snow cover in the warming treatment, which dramatically changes environmental conditions between warming and control treatments during winter. Given an eventual overload of the warming capacity by the heating cables, this could even lead to a cooling of warmed plots during winter time, which we did not intend to study here, and to strong changes in the soil hydrological properties during spring, when no snow can melt in warmed plots to sustain high soil water contents. Therefore, we only heated up the warming plots during the snow-free seasons, and thus we did not sample soils in winter months, where soils are not warmed. We now added specification in the Methods section (see below) and in Introduction section (Line 75) to clarify the warming treatment in the main text.

Manuscript:

The heating system is only running during the snow-free season (April to November) to avoid substantial differences in environmental conditions between warming and control treatments, due to warming-induced snow melting and adverse effects on soil hydrology (Lines 258-261).

The main findings reported (Table 1, Figure 1-2) are indeed interesting. However, it must be emphasised that all the reported results are based on correlations only. Therefore, greater caution is needed to describe these as correlations rather than causal relationships when inferring mechanisms in the discussion.

1-3. Author response:

We re-structured and rephrased the explanations to avoid over-interpreting the correlation results and added new citations to support the interpretations. Though the results are based on LMEs and on correlations, which is correct, the mechanisms underlying the observed changes have been published, e.g. on clay and Fe/Al oxide effects on P_i sorption, and are also hard to manipulate in the field.

Manuscript:

1-3-1. According to the principal component analysis (PCA), dithionite-extractable Fe was closely and positively associated with abiotic P immobilization (Fig. 2). Besides, dithionite-extractable Fe and crystalline Fe oxides were positively correlated with abiotic P immobilization and negatively correlated with (log-transformed) Olsen P (Fig. 3a-d). Iron oxyhydroxides have often been reported to act as very strong sorption sites for P_i in acidic soils, owing to their large specific surface area and positive surface charge, and this strong sorption capacity was reported to remain substantial at neutral soil pH (Boily et al., 2001; Frossard et al., 2005; Herndon et al., 2019; Memon 2008) Moreover, the sorption capacity of Fe oxyhydroxides increases with decreasing soil pH (Boily et al., 2001; Frossard et al., 2005; Memon 2008;

Nwoke et al., 2003). Thus, the lower soil pH in the warming treatment can further have enhanced abiotic P immobilization (Table 2), thereby restricting P_i availability in soil solution. Besides, the bonds between P_i and Fe oxyhydroxides are highly inert, implying a slower P return via desorption to the soil solution and reducing soil P availability within decadal warming (Frossard et al., 2005; Olander & Vitousek, 2004; Lines 132-145).

- 1-3-2. Soil texture and clay mineralogy are also relevant for abiotic P_i immobilization; especially clay minerals with a high specific surface area can form stable chemical bonds with P via calcium (Ca^{2+}) and magnesium (Mg^{2+}) bridges (Bünemann et al., 2010; Spohn 2020). In this study, sand content was significantly lower in warmed soil, whereas clay content was 11% higher than in control soil (though not significant). According to the results of PCA, abiotic P_i immobilization was negatively associated with soil sand content, while it co-varied positively with soil clay content (Fig. 2). Soils with higher sand content commonly exert lower P sorption capacity as the dominant quartz particles in the sand size fraction only have weak bonding affinity to P, whereas soils with high clay content often exhibit higher P sorption capacity, because of a large specific surface area and abundant bonding sites to P (Spohn 2020; Martin et al., 2018; Jalali and Jalali, 2016). Thereby, the lower sand content in combination with a higher clay content (but unaltered clay mineralogy) in warmed soil can be another reason for the intensified abiotic P_i immobilization (Lines 146-158).

Newly added references:

Herndon, E. M., Kinsman-Costello, L., Duroe, K. A., Mills, J., Kane, E. S., Sebestyen, S. D., ... & Wulfschleger, S. D. (2019). Iron (oxyhydr) oxides serve as phosphate traps in tundra and boreal peat soils. *Journal of Geophysical Research: Biogeosciences*, 124(2), 227-246.

Spohn, M. (2020). Phosphorus and carbon in soil particle size fractions: A synthesis. *Biogeochemistry*, 147(3), 225-242.

Jalali, M., & Jalali, M. (2016). Relation between various soil phosphorus extraction methods and sorption parameters in calcareous soils with different texture. *Science of the Total Environment*, 566, 1080-1093.

Specific comments:

Title: I'm not sure 'impairs' is an appropriate description of the results. Impairs means to diminish in function, which I don't think is reflected here in terms of P cycling. The manuscript shows interesting changes in P cycling, with immobilisation and increased enzyme investment, but I don't think this can be summarised as being an impaired.

1-4. **Author response:** We meant to express that long-term soil warming decreased (impaired) microbial P utilization, and greater production of phosphatase did not compensate for this, as microbial biomass P decreased. Now we changed the title to "Long-term soil warming decreases microbial phosphorus utilization by increasing abiotic phosphorus sorption and phosphorus losses" (Lines 1-2).

Line 34: 'By the end' instead of 'Until'

1-5. **Author response:** Done. Thank you (Line 38).

Lines 48-49: Sentence doesn't make sense to me - Please check language here ('did either promote or have no effects').

1-6. **Author response:** We replaced "did" with "can" to make the sentence clearer (Line 51).

Lines 51-53: Would be improved here by describing in more detail the studies have looked at P dynamics in warmed soils, and being more explicit about what is novel here. For example, recent paper on P mobilisation in warmed soils by Zhou et al: <https://doi.org/10.1111/gcb.15914>.

1-7. **Author response:** We thank the reviewer for this suggestion. We now added multiple references that investigate the warming effects on abiotic P processes, and more explicitly describe the novelty of this study.

Manuscript:

So far, a very limited number of warming studies have investigated soil abiotic P processes, and most of these studies assessed the warming effects on abiotic P processes indirectly, via comparing the changes of different P pools between warming and control treatments (Lie et al., 2022; Rui et al., 2012; Zhou et al., 2021). In this study, we applied a ³³P isotope pool dilution method (Wanek et al., 2019), which allows to quantify the process rates of gross and net (biotic and abiotic) P mobilization and immobilization, and thereby directly examined how specific soil P transformation processes responded to long-term soil warming. Moreover, soil P processes and P pools are highly connected and interacting (Bünemann et al., 2010; Frossard et al., 2005; Olander & Vitousek 2004; Reed et al., 2015; Sardans et al., 2006). Therefore, investigating soil P processes in warming studies can help to explain observed warming effects on soil and plant P pools, and to advance our understanding of soil P cycling processes in response to global warming (Lines 54-65).

Newly added references:

Lie, Z., Zhou, G., Huang, W., Kadowaki, K., Tissue, D. T., Yan, J., ... & Liu, J. (2022). Warming drives sustained plant phosphorus demand in a humid tropical forest. *Global Change Biology*.

Zhou, J., Li, X. L., Peng, F., Li, C., Lai, C., You, Q., ... & Lambers, H. (2021). Mobilization of soil phosphate after 8 years of warming is linked to plant phosphorus-acquisition strategies in an alpine meadow on the Qinghai-Tibetan Plateau. *Global Change Biology*, 27(24), 6578-6591.

Rui, Y., Wang, Y., Chen, C., Zhou, X., Wang, S., Xu, Z., ... & Luo, C. (2012). Warming and grazing increase mineralization of organic P in an alpine meadow ecosystem of Qinghai-Tibet Plateau, China. *Plant and Soil*, 357(1), 73-87.

Lines 59-60: The rationale behind studying changes across soil depth is very insubstantial. If soil depth is an important part of this study then more detail is needed on why responses in deep soils by differ, providing rationale for hypotheses on response differences across soil depths.

Lines 60-61: As above for soil depth, scarce information here on why seasonality is important in the context of this study, to provide a rationale for any hypotheses on seasonal dynamics. Suggest specifying what properties change seasonally and explicitly framing hypotheses on those seasonal changes (e.g. temperature/moisture – how change seasonally, how will this affect P dynamics under warming?).

Lines 63-64: Related to the above here it is stated that the overall objective of this study is to investigate how warming affects soil P... 'across soil depths and seasons'. Given this overall objective, please include more detail in the introduction on why depth/season is important (both for this study site – and in the context of other study sites).

1-8. **Author response:** We wrote a separate paragraph to explain why soil depth and seasonality are important in the context of this study. There are strong changes in soil physicochemistry and microbiology with soil depth, having considerable impacts on abiotic and biotic processes, including those related to soil P cycling, but those have rarely been studied in terms of global change effects. Moreover, several studies showed that warming effects can strongly change across seasons (see citations). However, since this study mainly focused on soil warming effects and most of the measured properties, pools, and processes had no interactions between warming, soil depth, and/or season, we prefer to keep the explanation brief.

Manuscript:

The warming response of soil P pools and soil P processes could be modulated by soil depth, since physicochemical soil properties, microbiological properties, and nutrient conditions strongly vary with soil depth (Lie et al., 2019; Xu et al., 2022; Zhou et al., 2021). Further variability in soil P cycling may arise from seasonal effects (Machmuller et al., 2016; Sardans et al., 2006; Lie et al., 2019; Zuccarini et al., 2020; Sistla & Schimel, 2013), e.g. seasonal changes in soil temperature and moisture, plant litter inputs, and rhizosphere processes. Thus, including soil depth and seasonality can provide more thorough results of warming effects on soil P cycling in this temperate forest, but also in other terrestrial systems (Lines 66-72).

Newly added references:

Machmuller, M. B., Mohan, J. E., Minucci, J. M., Phillips, C. A., & Wurzburger, N. (2016). Season, but not experimental warming, affects the activity and temperature sensitivity of extracellular enzymes. *Biogeochemistry*, 131(3), 255-265.

Sistla, S. A., & Schimel, J. P. (2013). Seasonal patterns of microbial extracellular enzyme activities in an arctic tundra soil: Identifying direct and indirect effects of long-term summer warming. *Soil Biology and Biochemistry*, 66, 119-129.

Lie, Z., Lin, W., Huang, W., Fang, X., Huang, C., Wu, T., ... & Liu, J. (2019). Warming changes soil N and P supplies in model tropical forests. *Biology and Fertility of Soils*, 55(7), 751-763.

Xu, M., Zhao, Z., Zhou, H., Ma, L., & Liu, X. (2022). Plant Allometric Growth Enhanced by the Change in Soil Stoichiometric Characteristics With Depth in an Alpine Meadow Under Climate Warming. *Frontiers in Plant Science*, 13, 860980-860980.

Line 74: The soil depths for this study are 0-10cm and 10-20cm. What soil horizons do these represent? I would have thought they are both representative of O or O-A horizons. Please include this information for these soils. Are soil properties across this 10cm depth differences sufficiently different to warrant using this as an experimental comparison? If there is a large difference, then it should be explained here so the reader understands that 0-10 cm vs. 10-20cm represents a valid comparison of different soil environments.

1-9. **Author response:**

We added Table 1 (see in the revised Display document), which shows the comparison of soil properties at 0-10 cm and 10-20 cm soil depth in the warming and control treatments. As you can see, many of these soil properties significantly differ at 0-10 cm and 10-20 cm depth, most of them decreasing. Besides, since the soil has developed from dolomite bedrock, it is in general very shallow, being Chromic Cambisols and Rendzic Leptosols, where at most spots in this forest one arrives at the C horizon at only 20-25 cm depth. We now replaced “topsoil” and “subsoil” with “0-10 cm” and “10-20 cm” throughout the manuscript to make the manuscript clearer, the previous topsoil referring to the A horizon, and the previous subsoil constituting the AC horizon.

Line 74: The seasonal comparison here is May, August, October; and includes no observation during the winter. I think some justification is needed to explain why these three months were selected, and when discussing seasonal dynamics it should be mentioned that no observations were made during the coldest months.

1-10. **Author response:**

As mentioned in 1-2, we added more explanation in the site description section in Methods (Lines 258-261), and also mentioned that no observations were made during the snow-free seasons in the Introduction (Line 75).

Line 80: Related to above comment, here the ‘10-20cm’ treatment is described as ‘subsoil’. I would suggest subsoil would come from the B horizon (usually much deeper, at least >30cm), and have significantly less organic matter than the top soil. Please present properties on these two soil horizon treatments, to provide a justification for their use as a treatment comparison.

1-11. **Author response:**

Thanks for the suggestion. As addressed in 1-2 and 1-9, we added a new table (Table 1) to provide more information on the differences of soil properties between these two soil depths. We haven’t measured soil organic matter content in our soils, but the soil organic carbon is almost double at 0-10 cm compared to 10-20 cm soil depth, indicating substantial differences between 0-10 cm and 10-20 cm soil depth at our site. Moreover, 0-10 cm represents the A horizon and 10-20 cm the AC horizons at this site. Almost all other physicochemical parameters also changed across these depth increments.

Line 83: Please describe how and why warming ‘facilitates plant P uptake from soil to reduce soil P pools’?

1-12. **Author response:**

We extended the explanation and added two meta-analysis papers to make it more understandable and convincing.

Manuscript:

Elevated soil temperatures, in general, facilitate plant growth, while warming had no effects on plant C:P ratios in temperate forest ecosystems (Lu et al., 2013; Yue et al., 2017). This indicates a higher plant P uptake, which removes P from soil to plant, in turn reducing total soil P pools (Lines 93-95).

Lines 87-91: Suggest review and re-write this text with greater focus. In its present form I find this text confusing and appears as a straw-man argument. Initially it is suggested that the results imply higher P demand and greater P uptake; but in the following sentence it is stated that root P contents did not change. Therefore, if root P contents did not change, isn't this evidence against greater P uptake?

1-13. **Author response:**

We modified the description to make it clearer.

Manuscript:

Elevated soil temperatures, in general, facilitate plant growth while warming had no effects on plant C:P ratios in temperate forest ecosystems (Lu et al., 2013; Yue et al., 2017). This indicates a higher plant P uptake, which removes P from soil to plant, in turn reducing total soil P pools. Kengdo et al., who studied fine root production, morphology, and element contents at the same site, reported an 128% increase in fine root production, 17% increase in fine root biomass, and unchanged fine root P content in the warming treatment compared to the control treatment. These results imply a higher P demand of the trees for fine root production and potentially greater root P uptake and P immobilization in long-lived plant biomass, ultimately reducing soil total P in warmed soils (Sardans et al., 2006; Lie et al., 2022; Zhou et al., 2021). However, this substantially higher fine root production compared to moderate biomass increase indicate a faster fine root turnover, which suggests a reflux of P from decomposing root necromass into the soil, attenuating the soil P loss pathway from plant uptake (Lines 93-105).

Newly added references:

Kwatocho Kengdo, S., Peršoh, D., Schindlbacher, A., Heinzle, J., Tian, Y., Wanek, W., & Borken, W. (2022). Long - term soil warming alters fine root dynamics and morphology, and their ectomycorrhizal fungal community in a temperate forest soil. *Global Change Biology*, 28(10), 3441-3458.

Lie, Z., Zhou, G., Huang, W., Kadowaki, K., Tissue, D. T., Yan, J., ... & Liu, J. (2022). Warming drives sustained plant phosphorus demand in a humid tropical forest. *Global Change Biology*.

Zhou, J., Li, X. L., Peng, F., Li, C., Lai, C., You, Q., ... & Lambers, H. (2021). Mobilization of soil phosphate after 8 years of warming is linked to plant phosphorus-acquisition strategies in an alpine meadow on the Qinghai-Tibetan Plateau. *Global Change Biology*, 27(24), 6578-6591.

Lines 92-106: A common problem in this manuscript is the lack of evidence drawn from the literature, to place the reported results in a wider context. This paragraph is one example of that. Many conjectures are made (e.g. lines 100-103) with no supporting literature cited.

1-14. **Author response:**

We now added several citations to support the explanation.

Manuscript:

Moreover, soil P can be lost from terrestrial systems as dissolved P and P that is bound to soil particles (particulate P), via leaching and erosion processes (Alewell et al., 2020; Bol et al., 2016;

Heathwaite & Dils, 2000; Reid et al., 2018; Sohrt et al., 2017). High precipitation at this forest site could have favored the leaching of dissolved organic and inorganic P (Bol et al., 2016; Sohrt et al., 2017), but this was not confirmed by soil solution chemistry (dissolved P in soil water was always below the detection limit; data not shown). Soil water P concentrations are generally low, but we found a decrease in 0.5 M NaHCO₃ extractable dissolved organic P (Olsen P_o) at 0-10 cm soil depth in the warming treatment, when calculating the ratio of Olsen P_o at 0-10 cm soil depth divided by Olsen P_o in 10-20 cm (decrease from 1.16 in controls to 0.99 in warmed soils), indicating a higher net downward transport (leaching) of Olsen P_o in warmed soils. Besides, even at high mean annual precipitation in old-growth forests, soil erosion and related particulate P losses likely do not play a large role, due to the high canopy and litter cover, effectively curtailing erosion in dense forests (Bavor 1956). Overall, net losses of total P from the soils were mainly due to downward leaching of Olsen P_o to lower soil layers, as the increased root P uptake was largely compensated by enhanced reflux of root P through increased root turnover (Lines 106-121).

Newly added references:

Alewell, C., Ringeval, B., Ballabio, C., Robinson, D. A., Panagos, P., & Borrelli, P. (2020). Global phosphorus shortage will be aggravated by soil erosion. *Nature communications*, 11(1), 1-12.

Heathwaite, A. L., & Dils, R. M. (2000). Characterising phosphorus loss in surface and subsurface hydrological pathways. *Science of the Total Environment*, 251, 523-538.

Sohrt, J., Lang, F., & Weiler, M. (2017). Quantifying components of the phosphorus cycle in temperate forests. *Wiley Interdisciplinary Reviews: Water*, 4(6), e1243.

Bol, R., Julich, D., Brödlin, D., Siemens, J., Kaiser, K., Dippold, M. A., ... & Hagedorn, F. (2016). Dissolved and colloidal phosphorus fluxes in forest ecosystems—an almost blind spot in ecosystem research. *Journal of Plant Nutrition and Soil Science*, 179(4), 425-438.

Baver, L.D., 1956. *Soil Physics*, third ed. John Wiley & Sons Inc., New York.

Lines 123-127: Here and elsewhere, greater caution is needed when inferring mechanisms based on correlations. (e.g. here a direct mechanism is inferred based on a correlation ‘increased Fe oxyhydroxide contents in warmed soils contributed to a greater abiotic P immobilisation strength’).

1-15. **Author response:**

(Please see Response 1-3).

Lines 140-150: This text about the geology of the study sites appears out of place here, perhaps this could be better placed in the methods or site description.

1-16. **Author response:**

Thanks for the suggestion! We now moved this text to the site description in the methods section (Lines 238-249).

Lines 150-160: This text needs a significant re-write to remove speculation and to base the discussion on the results and grounded in the wider literature. At present the text includes no references to other studies, but is full of speculation without any support. e.g. ‘the trend towards’.. ‘might have contributed,

but we cannot attributed this to a warming effect'. 'It seems rather unlikely that warming affected soil texture, but it may not be totally ruled out...'

It's fine to have some speculation, but it needs to be firmly grounded in a hypothesis based on evidence if not evidence from this study, then evidence from elsewhere).

1-17. **Author response:**

We modified this paragraph by minimizing speculations and adding multiple citations to make the discussion sounder.

Manuscript:

Soil texture and clay mineralogy are also relevant for abiotic P_i immobilization; especially clay minerals with a high specific surface area can form stable chemical bonds with P via calcium (Ca^{2+}) and magnesium (Mg^{2+}) bridges (Bünemann et al., 2010; Spohn 2020). In this study, sand content was significantly lower in warmed soil, whereas clay content was 11% higher than in control soil (though not significant). According to the results of PCA, abiotic P_i immobilization was negatively associated with soil sand content, while it co-varied positively with soil clay content (Fig. 2). Soils with higher sand content commonly exert lower P sorption capacity as the dominant quartz particles in the sand size fraction only have weak bonding affinity to P, whereas soils with high clay content often exhibit higher P sorption capacity, because of a large specific surface area and abundant bonding sites to P (Spohn 2020; Martin et al., 2018; Jalali and Jalali, 2016). Thereby, the lower sand content in combination with a higher clay content (but unaltered clay mineralogy) in warmed soil can be another reason for the intensified abiotic P_i immobilization.

So far, short-term (decadal) warming effects on soil metal oxides, soil texture and clay mineralogy have rarely been studied. Elevated temperatures favor weathering processes, increasing metal oxide contents in soils at decadal scales (Neupane et al., 2021; Qafoku 2015). Besides, higher temperatures can increase the soil clay content (Buol et al., 1990; Jenny 1994). However, these warming-induced changes are commonly believed to occur at longer time scales (Qafoku 2015; Buol et al., 1990; Jenny 1994). Comparatively, rapid (decadal) changes in soil metal oxides, soil texture and clay mineralogy have been shown in other studies in response to changes in management and fertilizer regime (Cornu et al., 2012; Fink et al., 2014; Liu et al., 2017; Mastro et al., 2020; Qafoku 2015). If such rapid changes could be triggered by soil warming as well, forest soil nutrient cycling could be affected substantially during the coming decades, and this will be the focus of further research at the site (Lines 146-168).

Newly added references:

Jalali, M., & Jalali, M. (2016). Relation between various soil phosphorus extraction methods and sorption parameters in calcareous soils with different texture. *Science of the Total Environment*, 566, 1080-1093.

Spohn, M. (2020). Phosphorus and carbon in soil particle size fractions: A synthesis. *Biogeochemistry*, 147(3), 225-242.

Liu, Y. L., Yao, S. H., Han, X. Z., Zhang, B., & Banwart, S. A. (2017). Soil mineralogy changes with different agricultural practices during 8-year soil development from the parent material of a Mollisol. *Advances in Agronomy*, 142, 143-179.

Fink, J. R., Inda, A. V., Almeida, J. A. D., Bissani, C. A., Giasson, E., & Nascimento, P. C. D. (2014). Chemical and mineralogical changes in a Brazilian Rhodic Paleudult under different land use and managements. *Revista Brasileira de Ciência do Solo*, 38, 1304-1314.

Buol, S. W., Sanchez, P. A., Weed, S.B., & Kimble, J.M. (1990). Predicted impact of climatic warming on soil properties and use. *Impact of carbon dioxide, trace gases, and climate change on global agriculture*, 53, 71-82.

Jenny, H. (1994). *Factors of soil formation: a system of quantitative pedology*. Courier Corporation.

Lines 177-191: This paragraph appears to summarise results from another study currently in review elsewhere. Overall, as I read this paper I'm confused as to whether it's a results paper or a review/summary paper (either are fine – but I would suggest the authors consider which to frame this as during revision).

1-18. **Author response:**

This manuscript is a result paper as one might see from the Methods section. We drew on this parallel study on microbial C-N-P limitation here to provide more evidence of that long-term warming-induced microbial P limitation. We now shortened the summary of their results.

Manuscript:

Soil microbes compete with abiotic cycling processes for available P in soil solution (Fig. 1; Reed et al., 2015; Olander & Vitousek, 2004). Reduced soil total P after long-term warming likely intensified this competition between abiotic immobilization and microbial processes in warmed soils. Moreover, enhanced abiotic Pi immobilization in the warming treatment depleted available P for soil microbes. In a C-N-P substrate addition experiment, Shi et al. (*Global Change Biology*, in revision) measured the stimulation of microbial growth in response to substrate amendments as an indicator of microbial element limitation in the same soils in August 2019. They found a strong stimulation of microbial growth in the combined C-P amendment than in the C-only addition in the warming treatment. In contrast, there was no difference in microbial growth stimulation between the combined C-P amendment and the C-only amendment in control soils. These results confirmed that, after 14 years of forest soil warming, soil microbes have increasingly become constrained by P in the warming treatment (Lines 182-194).

Lines 205-222: The conclusions paragraph is another example of the lack of depth here. The paragraph reads as a list of results and includes no references to other work.

1-19. **Author response:**

We shortened the results and modified the conclusion. In a conclusion, however, it is not usual to cite (multiple) references, but rather to summarize the results and give an outlook on the implications of the work to be published. We therefore remarked the importance of this study: 1)

providing novel data of long-term warming effects on soil P process rates; 2) long-term warming can drive temperate forest soils to become P limited; 3) which can further affect soil C and N cycling.

Manuscript:

In conclusion, this study contributes novel data on warming effects on soil P processes, providing a more comprehensive understanding of climate change effects on the soil P cycle (Fig. 1). Our results suggest that long-term warming strongly reduced soil available P, which resulted from substantial losses of TP (via leaching and plant P uptake) and enhanced Pi sorption (onto Fe oxide and clay minerals), and can turn temperate forest soils to P limited (Du et al., 2020). As a response, soil microbes changed their allocation of C (energy) and nutrients to acquire P (Reed 2015; Frossard et al., 2005; Bünemann et al., 2010), emphasizing the importance of investigating soil P cycling processes, together with C and N processes, in global warming research (Lines 214-222).

Newly added references:

Du, E., Terrer, C., Pellegrini, A. F., Ahlström, A., van Lissa, C. J., Zhao, X., ... & Jackson, R. B. (2020). Global patterns of terrestrial nitrogen and phosphorus limitation. *Nature Geoscience*, 13(3), 221-226.

Reviewer #2 (Remarks to the Author):

GENERAL COMMENTS:

This paper addresses an important topic in soil science: contributes to a better understanding of soil P biotic and abiotic cycling under main drivers of global change. In particular the work tries to shed some light on the effect of warming over abiotic processes (weathering, P sorption), processes that are very often overlooked on soil warming experiments (commonly centered in understanding biological responses).

A noteworthy result of this manuscript is the relation between the decrease of total soil P pools under warming with an increased sorption strength for inorganic P to iron oxyhydroxides and clay. This work proves that abiotic processes are important as biotic ones as drivers of important changes in nutrient cycling under global change. This work is of significance to the field and related fields. The processes described are not new (soil P pathways organic and inorganic cycling). However, the novelty of this work consists in applying a very complete set of measurements to a mid-term (14 years) warming experiment providing a complete in-depth study of soil P cycle.

This manuscript addresses topics of main-line interest to Nature Communications, it is well-written and thoroughly referenced. The title is appropriate, the abstract conveys the main points, and the overall conclusions are of wide interest. However, I have concerns on some points of the discussion and I miss a better graphical (Figures) support. If the points addressed bellow are improved enough I would recommend its publication by the requested journal.

I would ask the authors why they refer to dissolved inorganic/organic P instead of the commonly used “available P” or P-Olsen. DOP and DIP terms result confusing as they are mainly terms applied to water studies.

2-1. Author response:

Thanks for the suggestion. We replaced “dissolved inorganic/organic P” with “Olsen P_i/P_o” throughout the manuscript.

The authors affirm on page 4 (line 69) that inorganic P sorption decreases at higher temperatures and include a reference that support this statement. However, most literature demonstrate that sorption processes are higher at higher temperatures (when these are close to ambient temperatures 5-25°C).

2-2. Author response:

We apologize for the misuse of these references. In general, sorption is an exothermic process. Besides, increased temperature can decrease the attractive force between adsorbed molecules and the sorption surface. Thus, higher temperature should weaken the sorption process. We now updated the references.

Newly added references:

Hanyabui, E., Obeng Apori, S., Agyei Frimpong, K., Atiah, K., Abindaw, T., Ali, M., ... & Byalebeka, J. (2020). Phosphorus sorption in tropical soils.

From my point of view, it is missing in the text a better description of the soil type (classification, mineralogy, horizons depth) on the main text to provide elements for a more comprehensive discussion. At certain part of the discussion the authors refer to a high spatial heterogeneity in texture of the sampled soils and the presence of loamy moraine sediments as subsoils. This information would provide a better understanding of the site and soil processes if introduced before together with more accurate descriptions.

2-3. Author response:

Thanks for the suggestion! We added a separate paragraph describing the soil information in the site description section in the Methods.

Manuscript:

The bedrock is formed of dolomite. The soils are characterized as Chromic Cambisols and Rendzic Leptosols (based on the World Reference Base for Soil Resources), with high carbonate content and near-neutral pH. The 0-10 cm soil increment largely represents the A horizon, and the depth increment of 10-20 cm is an AC horizon. The depth increment below 20-25 cm is constituted by the C horizon, with more than 80% represented by coarse dolomite rock. Moreover, soils exhibit a high spatial heterogeneity in texture (primarily determined by micro-topography) and resemble a small-scale mosaic of autochthonous young A/C profile soils. The soils contain older loamy moraine sediments, which were deposited by glaciers during the last ice age, as subsoils. Besides, in the clay fraction, our site was dominated by kaolinites (~60%), followed by chlorites associated with a mixed layered clay (~30%), and illites (~10%). This high contents of kaolinite in the clay fraction, a very stable clay mineral with high to moderate Pi sorption capacity, clearly could not have formed during the last deglaciation phase (10,000 to 15,000 years). That indicates inputs via aeolian deposition from older nearby soil formations and inputs from the Central Alps. Accordingly, soil texture showed high spatial variability within the different control and warming plots (Lines 233-249).

The authors argue possible biotic pathways to explain the decrease of total P pools. This includes a higher uptake by plants explained by an increase in fine root production. What are the implications and limitations of the experimental design to evaluate plant response to warming? In contrast to what happens with current climate change, in this case the soil is warmed but not the aboveground which restricts or unbalances plant response to belowground adaptations.

2-4. Author response:

We agree with the reviewer that the results from soil warming experiment and soil+canopy warming experiment may differ, however, due to technical and financial constraints we are not aware of any warming experiment of a full mature temperate, boreal or tropical forest (above+belowground) with an average canopy height of >25 m. We would have ideas to do so, but this would also incur very strong unwanted micrometeorological effects and cost multi-millions of Euros to setup.

With canopy warming, it most likely can increase gross plant P uptake from soil due to facilitated photosynthesis, but can also increase litter P inputs to soils. However, due to the relocation of foliar P from old leaves to other tissues, especially in P restricted environments, soil+canopy warming experiment may have higher net plant P uptake than soil warming experiment. We are fully aware of

that we only warmed limited part of trees in each plot. This was why we toned down the interpretation of plant P uptake pathway in our manuscript (Lines 102-105 and Lines 118-121).

Higher Fe hydroxide content after warming treatments is attributed to faster weathering processes. This affirmation is surprising as a result of a 14-years warming. This is because weathering intensity directly relates to temperature, CO₂ concentration and precipitation but it is generally attributed to longer time scales (millennial). There aren't many published studies considering the role of weathering changes at decadal time scales. I would appreciate that the authors develop more this idea including more references. At the moment the effects of the climate-change induced changes in weathering are not well studied and understood.

2-5. Author response:

We agree with the reviewer that normally we would expect changes in soil metal oxide contents (as well as soil texture and mineralogy) would happen over longer time periods, and there are not many warming studies that have investigated this topic, given the common belief that soil texture and mineralogy does not change within 10-20 years. However, this is a belief and not grounded in science, and we collated all literature and cited it showing that global change (land use, elevated CO₂, warming) can change this at decadal scales. We were also surprised, but we show that this change happened. So this is not speculation, but grounded on robust data. We now wrote a separate paragraph, discussing the warming effects on soil metal oxides, soil texture and clay mineralogy. Due to the scarcity of other studies, we cited several examples of decadal changes of soil metal oxides, soil texture and mineralogy to soil management activities to strengthen the discussion. Besides, we will continue researching on this topic at our site in the future.

Manuscript:

So far, short-term (decadal) warming effects on soil metal oxides, soil texture and clay mineralogy have rarely been studied. Elevated temperatures favor weathering processes, increasing metal oxide contents in soils at decadal scales (Neupane et al., 2021; Qafoku 2015). Besides, higher temperatures can increase the soil clay content (Buol et al., 1990; Jenny 1994). However, these warming-induced changes are commonly believed to occur at longer time scales (Qafoku 2015; Buol et al., 1990; Jenny 1994). Comparatively, rapid (decadal) changes in soil metal oxides, soil texture and clay mineralogy have been shown in other studies in response to changes in management and fertilizer regime (Cornu et al., 2012; Fink et al., 2014; Liu et al., 2017; Mastro et al., 2020; Qafoku 2015). If such rapid changes could be triggered by soil warming as well, forest soil nutrient cycling could be affected substantially during the coming decades, and this will be the focus of further research at the site (Lines 159-168).

Newly added references:

Buol, S. W., Sanchez, P. A., Weed, S.B., & Kimble, J.M. (1990). Predicted impact of climatic warming on soil properties and use. *Impact of carbon dioxide, trace gases, and climate change on global agriculture*, 53, 71-82.

Jenny, H. (1994). *Factors of soil formation: a system of quantitative pedology*. Courier Corporation.

Liu, Y. L., Yao, S. H., Han, X. Z., Zhang, B., & Banwart, S. A. (2017). Soil mineralogy changes with different agricultural practices during 8-year soil development from the parent material of a Mollisol. *Advances in Agronomy*, 142, 143-179.

Fink, J. R., Inda, A. V., Almeida, J. A. D., Bissani, C. A., Giasson, E., & Nascimento, P. C. D. (2014). Chemical and mineralogical changes in a Brazilian Rhodic Paleudult under different land use and managements. *Revista Brasileira de Ciência do Solo*, 38, 1304-1314.

Apart from clay mineralogy, what are the primary minerals? Which is the mineral phase attributed to the crystalline Iron? When compared with the total mineralogy (not only clays), is it possible to estimate the % of amorphous and crystalline Iron?

2-6. Author response:

The primary mineral is dolomite at our site. Goethite (iron hydroxide; α -FeO(OH)) constitutes the main phase of crystalline Fe in all soil samples at the Achenkirch site. We determined the content of crystalline iron and of amorphous iron in soils by wet chemistry (dithionite- and oxalate-extractable Fe). Oxalate-extractable represents the amorphous Fe oxyhydroxides (plus some humic-bound Fe) and dithionite-extractable minus oxalate-extractable Fe is the crystalline Fe. For the determination of soil mineralogy, we used X-ray diffraction (XRD), which only detects crystalline phases such as goethite, lepidocrocite, pyrite, hematite, etc. Ferrihydrite is mostly amorphous or very poorly crystallized. It is an Fe mineral, which can hardly be detected by XRD. Based on the results of Fe oxide contents, we get a range for the crystallised iron mineral (goethite) from 0.2 to 0.6 mass%, and a range for the amorphous iron (ferrihydrite) from 0.07 mass% to 0.19 mass%. These results also fit quite well to the results by XRD, where we found goethite in traces, which means <1 mass%.

Figures can be improved. Figure 1 can incorporate the complete variable name on PCA loadings. It is recommendable to use color blind friendly palettes (easy available in R). More elements for discussion could be derived from Figure 1, as the meanings or interpretation for PC1 or PC2.

2-7. Author response:

We updated our PCA figure with complete variable names and a color-blind friendly palette (the palette is from: <http://jfly.iam.u-tokyo.ac.jp/color/>; please see Figure 2 in the revised Display document).

On Figure 2, samples could be distinguished among treatments. I recommend the authors to include a third and final Figure with a schematic representation of the main processes described. I would consider this third Figure essential to achieve publication in the requested journal.

2-8. Author response:

Thank you for your suggestion. We now use different symbols for warming (triangle) and control (circle) treatments in Figure 3. Besides, we made a schematic illustration (Figure 1), visualizing our results of long-term soil warming effects on soil P cycling (please see the revised Display document).

I encourage the authors to reinforce in a clearer way what is new from their contribution, and to remark its importance in relation to the global understanding of soil carbon and nutrient cycling.

2-9. Author response:

Thanks for the suggestion. We shortened the results summary and remarked the importance: 1) providing novel data of long-term warming effects on soil P process rates; 2) long-term warming can drive temperate forest soils to become P limited; 3) which can further affect soil C and N cycling.

Manuscript:

In conclusion, this study contributes novel data on warming effects on soil P processes, providing a more comprehensive understanding of climate change effects on the soil P cycle (Fig. 1). Our results suggest that long-term warming strongly reduced soil available P, which resulted from substantial losses of TP (via leaching and plant P uptake) and enhanced Pi sorption (onto Fe oxide and clay minerals), and can turn temperate forest soils to P limited (Du et al., 2020). As a response, soil microbes changed their allocation of C (energy) and nutrients to acquire P (Reed et al., 2015; Frossard et al., 2005; Bünemann et al., 2010), emphasizing the importance of investigating soil P cycling processes, together with C and N processes, in global warming research (Lines 214-222).

Newly added references:

Du, E., Terrer, C., Pellegrini, A. F., Ahlström, A., van Lissa, C. J., Zhao, X., ... & Jackson, R. B. (2020). Global patterns of terrestrial nitrogen and phosphorus limitation. *Nature Geoscience*, 13(3), 221-226.

SPECIFIC COMMENTS

Pag 3 line 35: current warming is already above 1°C above pre-industrial levels

2-10. **Author response:**

We modified the sentence into “By the end of the century, global atmospheric temperatures are predicted to increase up to 5.7 °C above pre-industrial levels.” (Lines 38-39).

Pag 5 line 100: Soil water loss is expected to be higher at the top soil than at the subsoil, is it like this? Can this relate to changes on DOP and DIP between different depths through a higher reduction of diffusion capacity on the top soil?

2-11. **Author response:**

This is an interesting notion, but we found similar responses of soil water content at both soil depths. Usually warming causes a soil drying effect, which, in this high precipitation area, is always reset after frequent rainfall. Topsoils and subsoils were therefore similarly affected. Moreover, according to the results of soil water content, 0-10 cm soil depths had higher soil water contents than the 10-20 cm layers (Table 1), which is likely due to a significantly lower soil sand content at 0-10 cm depth, implying potentially less soil water losses at 0-10 cm than at 10-20 cm depth. Besides, we modified the sentence to make this clearer (Line 115).

Pag 7 line 139: First time mention to elements of soil description, better mention them earlier.

2-12. **Author response:**

Thanks for the suggestion. As mentioned in 2-3, we moved this description to the site description section (in the Methods section; Lines 250-259) as this information fits better in that section. By doing so, if readers are interested in the soil formation and composition of our site, they can easily target where to find this information.

Pag 7 line 144: better “moderate” instead of high P_i sorption capacity when referring to kaolinite. It has lower sorption capacity than 2:1 clay. The same for line 175 when is mentioned “strong P-sorbing” kaolinite.

2-13. **Author response:**

We replaced “strong” with “moderate”. Thank you.

Pag 8 line 165: In relation with a previous comment asking for more information on mineralogy, are there proves of apatite precipitation on a general diffractogram?

2-14. **Author response:**

Thank you for this comment! Based on the XRD diffractograms, apatite was detectable in the Achenkirch soils, but only in small amounts being non-quantifiable. We revised the manuscript accordingly.

Manuscript:

The forest soils studied here have very high contents in exchangeable Ca^{2+} and Mg^{2+} , the cation-exchange capacity being high, with a base saturation close to 100% (Table 1). Considering the near-neutral soil pH (Table 1), Ca^{2+} could play a major role in P_i precipitation in the form of Ca-apatite, which is highly insoluble and therefore might act as another geochemical sink [$Ca_{10}(PO_4)_6(OH)_2$] (Tunesi et al., 1999). However, according to the X-ray diffractograms, the apatite content was very low at our site. Besides, long-term warming did not affect the exchangeable Ca^{2+} pool (Table 2). Thus, the potential for Ca^{2+} -P precipitation does not play a large role in both, the warming and control treatments, and it did not cause the higher abiotic P immobilization in the warming treatment (Fig. 2). Overall, this increased abiotic P immobilization in warmed soils mainly resulted from increased P sorption onto Fe oxyhydroxides and clays (dominated by moderate P-sorbing kaolinite; Lines 169-180).

Pag, first paragraph: Could you discuss or compare the effect of inorganic P sorption vs. the increased root uptake when considering P limitation for the microbial pool? (a limitation that in turn may explain the microbial pool decrease under warming).

2-15. **Author response:**

We made two additional Pearson correlation figures (Fig. 3e and 3f) to show the relationships between microbial biomass P with total soil P and total Fe oxide content. Besides, we now briefly discuss how reduced total soil P and increased sorption onto Fe oxide potentially affect microbial P biomass pool.

Manuscript: This diminished P availability, due to decreased TP and increased P sorption, can further decrease microbial biomass P (MBP; Table 2; Fan et al., 2021; Joergensen et al. 1995). According to the results of PCA and Pearson coefficient correlations, MBP is tightly and positively associated with total soil P, while it has weaker but significant negative relations with total Fe oxide (Fig. 2, 3e, and 3f), implying stronger effects on MBP limitations from TP losses than from increased sorption (onto Fe oxide; Lines 194-200).

Newly added references:

Fan, Y., Lu, S., He, M., Yang, L., Hu, W., Yang, Z., ... & Yang, Y. (2021). Long-term throughfall exclusion decreases soil organic phosphorus associated with reduced plant roots and soil microbial biomass in a subtropical forest. *Geoderma*, 404, 115309.

Joergensen, R. G., Kübler, H., Meyer, B., & Wolters, V. (1995). Microbial biomass phosphorus in soils of beech (*Fagus sylvatica* L.) forests. *Biology and Fertility of Soils*, 19(2), 215-219.

Methodology: Please specify why the sampling design was performed at 0-10 and 10-20 and how relates with soil horizons and composition.

2-16. **Author response:**

We added a new table (Table 1, please see in the revised Display document) showing the comparison of soil properties from these two soil depths in the warming and control treatments.

Please specify protocol or references for chloroform fumigation-extraction applied (line 287).

2-17. **Author response:**

Thanks for reminding us of this. We added the reference (Line 293 and Lines 584-585, reference 62).

REVIEWER COMMENTS

Reviewer #3 (Remarks to the Author):

Major comments

I think that the authors have responded to all the questions made by the two previous referees leaving clear all, but less the question of the increase of P leaching under warming, (see comments in the response letter) in my mind. I think that previous to the publication authors should allocate a further effort in leave clearer why and how warming increase leaching of P. We must take into account two circumstances that not should help to increase P leaching under warming. First the fact of that P is scarcely soluble, and second that warming increase P fixation in Fe oxides (by dropping pH) and plant P uptake thus taking off chance to increases P leaching. Thus, I believe that the authors should to provide a more consistent explanation of why warming increase the amounts of P leached out the soil. He seem to argue a higher root turn over leaving more P able to by leached, but the writing of this part is not very clear. In this sense some parts of the manuscripts make exrange this argument of more P losses by high root turn over and sounds partially contradictory:

Lines 96-104: "...reported an 128% increase in fine root production, 17% increase in fine root biomass, and unchanged fine root P content in the warming treatment compared to the control treatment. These results imply a higher P demand of the trees for fine root production and potentially greater root P uptake and P immobilization in long-lived plant biomass, ultimately reducing soil total P in warmed soils^{15,19,21}. this substantially higher fine root production compared to moderate biomass increase indicate a faster fine root turnover, which suggests a reflux of P from decomposing root necromass into the soil, attenuating the soil P loss pathway from plant uptake."

Lines 113-121 "Olsen Po in 10-20 cm (decrease from 1.16 in controls to 0.99 in warmed soils), indicating a higher net downward transport (leaching) of Olsen Po in warmed soils. Besides, even at high mean annual precipitation in old-growth forests, soil erosion and related particulate P losses likely do not play a large role, due to the high canopy and litter cover, effectively curtailing erosion in dense forests³⁸. Overall, net losses of total P from the soils were mainly due to downward leaching of Olsen Po to lower soil layers, as the increased root P uptake was largely compensated by enhanced reflux of root P through increased root turnover.

This is not sufficient clear are the authors claiming that in warming plots the higher P-Olsen leaching is due to the higher Phosphate release from dye roots by the higher root biomass with faster turnover, supposing that more Olsen-P is free in soil to be leached?.

But this argument is not full consistent given that first Phosphate has low solubility and moreover phosphate is also more uptaken by plants and fixed in Fe salts!. The authors can close better this part of the manuscript results, apart form this the manuscript is now suitable after a minor revision to be published in Nature Communications

Minor comments

Lines 29-30. "Warming decelerated the gross rates of phosphate mobilization by 21%, likely due to decreased soil total P pools (substrates)" What here means mobilization? Leaching? Release by mineralization?.

Line 219. "temperate forest soils to become P limited⁵⁹" Moreover this is in the end Results/discussion paragraph that should be a synthesis of this study and the cite referring to other study provide some confusion. Thus I advise to make a concrete reference of the pass studies. For

example “Thus warming reduced P-availability..... in this study consistent with previous studies suggesting temperate forest soils under warming can become P limited⁵⁹”

I advise a minor revision previous to publication

Reviewer #4 (Remarks to the Author):

Tian et al. revised their manuscript following the comments of two reviewers. In the revised manuscript, they (i) extended the method descriptions, (ii) included a new figure and a new table, and (iii) rewrote parts of the manuscript. Altogether, the revision was done very thoroughly and most concerns of the reviewers were resolved. Yet, the introduction and hypotheses need to be improved and a broader perspective need to be implemented still. Because some of the findings challenge textbook knowledge, the authors must put more effort in eliminating alternative explanations. Finally, some minor issues should be incorporated before the study can be published. In the following, the original reviewer comments are given in orange, the authors' response in green and my new comments in black.

Comments on author responses to reviewers

Reviewer #1: In its current format the manuscript fails to provide sufficient background to provide a rationale for the presented hypotheses

Author response (shortened): We wrote a separate paragraph to explain why soil depth and seasonality are important in the context of this study.

New reviewer: There is still a mismatch between the background provided in the introduction and the hypotheses. On the one hand, neither soil depth nor seasonality (newly introduced) form part of the hypotheses. On the other hand, the expected directions of the warming effects ("promote", "increase", "decrease") are not backed up by explanations in the preceding introduction. For example, lines 49-54 do not explain responses of P cycling that are later stated as hypotheses. Therefore, the structure of the introduction and its match with the hypotheses need to be improved.

Reviewer #1: Lines 205-222: The conclusions paragraph is another example of the lack of depth here. The paragraph reads as a list of results and includes no references to other work.

Author response: We shortened the results and modified the conclusion. In a conclusion, however, it is not usual to cite (multiple) references, but rather to summarize the results and give an outlook on the implications of the work to be published. We therefore remarked the importance of this study: 1) providing novel data of long-term warming effects on soil P process rates; 2) long-term warming can drive temperate forest soils to become P limited; 3) which can further affect soil C and N cycling.

New reviewer: In addition to my comment on the phrasing (see below), I am still not convinced by the broader context of the conclusion. Why should we worry about soil (micro)organisms whose growth is potentially limited by P? And in which way would this be linked to soil C and N cycling – again stressing the relevance of this link? Could P-limited soil (micro)organisms have an impact on the C sink function of the soil? Could P-limited soil (micro)organisms influence N losses (open N cycle if N is no longer limiting)? I strongly recommend that the authors spend some more thoughts on the broader implications of their study.

Reviewer #2: The authors argue possible biotic pathways to explain the decrease of total P pools. This includes a higher uptake by plants explained by an increase in fine root production. What are the implications and limitations of the experimental design to evaluate plant response to warming? In contrast to what happens with current climate change, in this case the soil is warmed but not the aboveground which restricts or unbalances plant response to belowground adaptations.

Author response: We agree with the reviewer that the results from soil warming experiment and soil+canopy warming experiment may differ, however, due to technical and financial constraints we are not aware of any warming experiment of a full mature temperate, boreal or tropical forest (above+belowground) with an average canopy height of >25 m. We would have ideas to do so, but this would also incur very strong unwanted micrometeorological effects and cost multi-millions of Euros to setup. With canopy warming, it most likely can increase gross plant P uptake from soil due to facilitated photosynthesis, but can also increase litter P inputs to soils. However, due to the relocation of foliar P from old leaves to other tissues, especially in P restricted environments, soil+canopy warming experiment may have higher net plant P uptake than soil warming experiment. We are fully aware of that we only warmed limited part of trees in each plot. This was why we toned down the interpretation of plant P uptake pathway in our manuscript (Lines 102-105 and Lines 118-121).

New reviewer: I fully agree with the arguments of the authors. However, this constraint needs to be discussed briefly in the main text of the manuscript. If this explanation is missing, the reader will be tempted to infer implications e.g., on growth limitation of the forest by P and its consequences for the forest C sink, that must not be drawn based on the experimental design.

Reviewer #2: Higher Fe hydroxide content after warming treatments is attributed to faster weathering processes. This affirmation is surprising as a result of a 14-years warming. This is because weathering intensity directly relates to temperature, CO₂ concentration and precipitation but it is generally attributed to longer time scales (millennial). There aren't many published studies considering the role of weathering changes at decadal time scales. I would appreciate that the authors develop more this idea including more references. At the moment the effects of the climate-change induced changes in weathering are not well studied and understood.

Author response: We agree with the reviewer that normally we would expect changes in soil metal oxide contents (as well as soil texture and mineralogy) would happen over longer time periods, and there are not many warming studies that have investigated this topic, given the common belief that soil texture and mineralogy does not change within 10-20 years. However, this is a belief and not grounded in science, and we collated all literature and cited it showing that global change (land use, elevated CO₂, warming) can change this at decadal scales. We were also surprised, but we show that this change happened. So this is not speculation, but grounded on robust data. We now wrote a separate paragraph, discussing the warming effects on soil metal oxides, soil texture and clay mineralogy. Due to the scarcity of other studies, we cited several examples of decadal changes of soil metal oxides, soil texture and mineralogy to soil management activities to strengthen the discussion. Besides, we will continue researching on this topic at our site in the future.

New reviewer: With due respect, the time scales associated with weathering are no belief but itself also corroborated by data (see body of literature on changes in soil properties in chronosequence studies). Nevertheless, I am in favor of challenging the long-term perspective of weathering. However, this can only be achieved by thoroughly eliminating alternative explanations for your findings. One such explanation is an a priori difference between the pairs of plots. For example, the high heterogeneity within the control and warming treatments (e.g. l. 238-239) increases the risk of initially selecting control and warming plots differing in soil properties – just by chance. These concerns can be dispelled by data on soil properties of the plots at the time of the experimental set up i.e., before the start of the warming treatment. Are data on initial soil properties such as soil texture and mineralogy available? Or do archived soil samples exist that could be analyzed still? How do these data relate to your findings 14 years later? Without data on initial soil properties, the

causality of the warming effects on soil texture and mineralogy remains questionable and all corresponding statements need to be toned down.

Reviewer #2: Apart from clay mineralogy, what are the primary minerals? Which is the mineral phase attributed to the crystalline Iron? When compared with the total mineralogy (not only clays), is it possible to estimate the % of amorphous and crystalline Iron?

Author response: The primary mineral is dolomite at our site. Goethite (iron hydroxide; α -FeO(OH)) constitutes the main phase of crystalline Fe in all soil samples at the Achenkirch site. We determined the content of crystalline iron and of amorphous iron in soils by wet chemistry (dithionite- and oxalate-extractable Fe). Oxalate-extractable represents the amorphous Fe oxyhydroxides (plus some humic-bound Fe) and dithionite-extractable minus oxalate-extractable Fe is the crystalline Fe. For the determination of soil mineralogy, we used X-ray diffraction (XRD), which only detects crystalline phases such as goethite, lepidocrocite, pyrite, hematite, etc. Ferrihydrite is mostly amorphous or very poorly crystallized. It is an Fe mineral, which can hardly be detected by XRD. Based on the results of Fe oxide contents, we get a range for the crystallised iron mineral (goethite) from 0.2 to 0.6 mass%, and a range for the amorphous iron (ferrihydrite) from 0.07 mass% to 0.19 mass%. These results also fit quite well to the results by XRD, where we found goethite in traces, which means <1 mass%.

New reviewer: The authors should include this description in the methods' part of the manuscript.

Additional comments by new reviewer:

36-37 The conclusive sentence is too broad. Link back to climate change, what would this mean for the forest? P limitation = detrimental effects on growth?

45-48 and 54-65 redundant, please merge.

80 no mention of references within hypotheses (should be explained already in the introduction)

98 "content" is ill defined. Do you refer to stocks or concentrations?

109-110 Unclear: does "dissolved P in soil water" refer to DIP, DOP or TDP concentrations?

115, 119 Awkward phrase "leaching of Olsen-Po": Here you interpret your results and thus, go beyond your own measurements. Replace by "leaching of Po" (or DOP if you consider the dissolved phase most relevant)

118-121 Statements too strong because you did neither determine DOP leaching nor root P uptake/reflux, please tone down. "net losses of total P from the soils seem to be mainly due to" and "as the increased root P uptake likely was largely compensated".

173-174 Where are these results shown/specified?

177 Statement too strong, please rephrase ("it cannot explain the higher abiotic P immobilization").

189 "stronger stimulation" (because "than" follows in the second part of the sentence)

217 "(likely via leaching and plant P uptake)"

198/200/218 “Fe oxide” has not been introduced. Up to here (l. 198), you referred to Fe oxyhydroxides/dithionite-/oxalate-extractable Fe. Please improve terminology and the explanations on the different fractions when mentioned first.

216-217 Awkward phrase “soil available P”. Commonly, terms such as “plant-available/bioavailable P” in soil are used. Please rephrase.

218-219 Awkward phrase “can turn temperate forest soils to P limited”. By definition, the growth of organisms – but not the soil – can be limited by nutrients. Please rephrase.

234 Add proper citation for soil classification!

244 “The high contents of kaolinite”

291-301 This description is confusing, for Olsen P just NaHCO₃ is used as extractant. However, the authors state that they used persulfate digestion in addition (l. 291-292). Please clarify how the Olsen P method was embedded (?) in the procedure used for the determination of MBP.

332 Why Fe oxides? There are also crystalline Fe oxyhydroxides.

Table 1:

- Please add the number of replicates (n = 6 for each treatment) in the caption. However, I do not get why some measurements total 18 while others sum up to 6 (see footnote). Were pseudoreplicates used in the former case?
- What does d.s. mean? Dry soil? Please explain in caption.
- Add different letters (a, b) for all cases in which the control and the warming plots differed significantly.

Figure 1: You did not measure plant P (= either above- or belowground P stocks or both). Stick to those variables for which results are presented in this manuscript.

Fig. 3: The calculation of crystalline Fe includes dithionite extracted P. Therefore, show only one (delete either subfigures a+b or c+d).

We thank the reviewers for their careful reading and constructive comments, which greatly helped us to further improve our manuscript! We addressed each comment individually. Given that the new reviewer #4 commented on the former comments of reviewer #1 and #2 and our responses, we adopted the coloring scheme. **The comments of the former reviewers #1 and #2 are in Orange/Red, our previous responses to reviewers #1 and #2 in Green. The comments by the new reviewers #3 and #4 are in Black, our responses to reviewers #3 and #4 are in Blue** and our changes in the revised manuscript are highlighted in yellow and indexed with the line numbers in the revised manuscript. Besides, to make the story coherent and smooth, we did minor revisions in the sections, “Warming effects on abiotic P processes” and “Warming effects on biotic P processes”, and highlighted these changes in Green in the revised manuscript.

Reviewer #3 (Remarks to the Author):

Major comments

I think that the authors have responded to all the questions made by the two previous referees leaving clear all, but less the question of the increase of P leaching under warming, (see comments in the response letter) in my mind. I think that previous to the publication authors should allocate a further effort in leave clearer why and how warming increase leaching of P. We must take into account two circumstances that not should help to increase P leaching under warming. First the fact of that P is scarcely soluble, and second that warming increase P fixation in Fe oxides (by dropping pH) and plant P uptake thus taking off chance to increases P leaching. Thus, I believe that the authors should to provide a more consistent explanation of why warming increase the amounts of P leached out the soil. He seems to argue a higher root turn over leaving more P able to by leached, but the writing of this part is not very clear. In this sense some parts of the manuscripts make exrange this argument of more P losses by high root turn over and sounds partially contradictory:

Lines 96-104: “ ...reported an 128% increase in fine root production, 17% increase in fine root biomass, and unchanged fine root P content in the warming treatment compared to the control treatment. These results imply a higher P demand of the trees for fine root production and potentially greater root P uptake and P immobilization in long-lived plant biomass, ultimately reducing soil total P in warmed soils^{15,19,21}. this substantially higher fine root production compared to moderate biomass increase indicate a faster fine root turnover, which suggests a reflux of P from decomposing root necromass into the soil, attenuating the soil P loss pathway from plant uptake.”

Lines 113-121 “Olsen Po in 10-20 cm (decrease from 1.16 in controls to 0.99 in warmed soils), indicating a higher net downward transport (leaching) of Olsen Po in warmed soils. Besides, even at high mean annual precipitation in old-growth forests, soil erosion and related particulate P losses likely do not play a large role, due to the high canopy and litter cover, effectively curtailing erosion in dense forests³⁸. Overall, net losses of total P from the soils were mainly due to downward leaching of Olsen Po to lower soil layers, as the increased root P uptake was largely compensated by enhanced reflux of root P through increased root turnover.

This is not sufficient clear are the authors claiming that in warming plots the higher P-Olsen leaching is due to the higher Phosphate release from dye roots by the higher root biomass with faster turn-over, supposing that more Olsen-P is free in soil to be leached?.

But this argument is not full consistent given that first Phosphate has low solubility and moreover phosphate is also more uptaken by plants and fixed in Fe salts!. The authors can close better this part of the manuscript results, apart from this the manuscript is now suitable after a minor revision to be published in Nature Communications

3-1. Author response:

Thank you for this comment! We re-evaluated different potential pathways for soil P losses, i.e., plant P uptake, downward P transportation (both dissolved and particulate P), atmospheric P losses, and P losses via soil erosion/runoff. We re-wrote the section, “Warming effects on soil P pools and P losses”, discussing the potential contributions of each pathway in our revised manuscript.

Instead of considering dissolved organic P leaching as the main reason for the total soil P losses, we here more specifically expand on other factors already mentioned in the previous manuscript version, i.e., (1) plant P uptake (experimental setup induced net P transfer from soil to plant) and (2) downward P transportation (especially particulate P through preferential flow paths) as the main reasons for total soil P losses at our site, whereas P losses to the atmosphere and through soil erosion (likely) played minor roles (for a detailed description, please read the revised manuscript below and in lines 90-150). Besides, we accordingly updated Fig.1, the schematic illustration, and linked with the section, “Warming effects on abiotic processes” in lines 211-215.

Revised manuscript:

After 14 years of warming, we found substantial losses of total soil P (TP) from the 0-10 cm (-18%) and the 10-20 cm (-30%) soil layer in the warming treatment compared to the control treatment (Tables 1 and 2). There are different pathways potentially causing soil TP losses, i.e., plant P uptake, downward transportation of particulate and dissolved P, atmospheric P losses, and P losses through soil erosion/runoff (Fig. 1).

Elevated temperatures, in general, facilitate plant growth while warming had no effects on plant C:P ratios in temperate forest ecosystems (Yue et al., 2017; Lu et al., 2013). Greater biomass production indicates higher plant P uptake, which removes P from soil and allocates this to the plant compartment, in turn reducing the total soil P pools. Kengdo et al., (2022) who studied fine root production, morphology, and element contents at the same site, reported a 128% increase in fine root production and a 17% increase in fine root biomass, but unaltered fine root P concentrations in the warming treatment compared to the control treatment. These results imply a higher P demand of the trees for fine root production, but they do not account for potentially greater plant P immobilization in aboveground tissues (Lie et al., 2022). The potentially larger aboveground P stock should eventually return to the soils, which might (partially) compensate the soil P losses by fine root uptake and aboveground P allocation in the warming treatment (Sohrt et al., 2017). However, due to the small plot size (2 by 2 meters) and soil-only warming (without canopy warming), the roots and canopies of trees are shared across the warming and the control treatments, and across experimental and non-experimental areas, causing comparable litterfall P returns in warmed and un-warmed areas. Therefore, increased gross plant P uptake in the warming treatment combined with comparable plant litter P returns between treatments will result in the redistribution of P, ultimately reducing soil TP in warmed soils in the long run.

Soil P can also be lost from terrestrial systems as dissolved P and P that is bound to soil particles (e.g. colloid-associated P and particulate P) through subsurface flux (Sohrt et al., 2017; Bol et al., 2016). High precipitation at this forest site (~1500 mm MAP and ~800 mm seepage) could have favored the leaching of dissolved P (Sohrt et al., 2017; Bol et al., 2016), but this was not confirmed by soil solution chemistry (dissolved inorganic P in soil water was always below the detection limit at ~1 μM ; data not shown). Colloidal particles (1-1000 nm size range) have a high specific surface area, are typically composed of organic matter and metal oxyhydroxides, and thus can immobilize and transport a considerable amount of organic and inorganic P (Missong et al., 2018). Moreover, colloid-associated P contributes up to 91% of total subsurface P fluxes in forest soils (Missong et al., 2018). At our site, warming increased fine root production (+128%) more than fine root biomass (+17%), which implies a fast root turnover in warmed soil (Kengdo et al., 2022). When roots die, their previous volume expansion can be preserved via the formation of macropores that might further generate effective and long-lasting preferential flow paths in soils, especially in undisturbed forests (Sohrt et al., 2017; Bol et al., 2016). Preferential flow paths have been reported to significantly enhance colloidal and (particulate) dissolved P transport, due to fast transit times and therefore

reduced interaction with the soil matrix (Sohrt et al., 2017; Bol et al., 2016; Missong et al., 2018). Hereby, high precipitation and seepage at this forest site and extension of root-induced preferential flow paths in warmed soils could have favored larger amounts of dissolved and colloid-associated P to migrate to deep soil layers, causing enhanced soil P losses in the warming treatment (Sohrt et al., 2017; Missong et al., 2018).

Moreover, subsurface P losses might be promoted by soil warming, as desorption processes increase stronger than sorption processes at higher temperatures (Conant et al., 2011; Hanyabui et al., 2020; Yuan et al., 2015). Phosphate sorption shows little temperature sensitivity, with Q_{10} values (Bai et al., 2017; Amarakoon et al., 2019) between 1.0 and 1.1, while desorption reactions of inorganic and organic matter have Q_{10} values of 1.2-2.0 (Conant et al., 2011). This may not only increase dissolved P losses, but also colloidal and particulate ones due to sorption of P_i to colloidal and nano-particulate clays and metal oxyhydroxides, the latter of which significantly increased in warmed soils.

Direct soil P losses to the atmosphere are negligible, with phosphine being the only gaseous P form. Phosphine is highly reduced and reactive, being rapidly oxidized to non-gaseous forms of P under aerobic conditions, hence explaining the very low gaseous P emissions from soils (Kehler et al., 2021). Substantial soil P losses can occur by wind erosion and by water erosion (Sohrt et al., 2017; Bol et al., 2016), but erosional losses are likely not affected by warming in this experiment. Overall, (greater) net losses of TP from the warmed soils resulted from multiple processes, including increased net plant P uptake and subsurface dissolved and colloidal/particulate P fluxes, while atmospheric and erosional losses played minor roles.

Newly added references:

Missong, A., Holzmann, S., Bol, R., Nischwitz, V., Puhmann, H., Wilpert, K. V., ... & Klumpp, E. (2018). Leaching of natural colloids from forest topsoils and their relevance for phosphorus mobility. *Science of the Total Environment*, 634, 305-315.

Bai, J., Ye, X., Jia, J., Zhang, G., Zhao, Q., Cui, B., & Liu, X. (2017). Phosphorus sorption-desorption and effects of temperature, pH and salinity on phosphorus sorption in marsh soils from coastal wetlands with different flooding conditions. *Chemosphere*, 188, 677-688.

Amarakoon, I., Zvomuya, F., & Motaung, M. L. (2019). Temperature-dependency of phosphorus sorption by Goethites and tropical soils amended with woodchip biochar. *Agrosystems, Geosciences & Environment*, 2(1), 1-6.

Kehler, A., Haygarth, P., Tamburini, F., & Blackwell, M. (2021). Cycling of reduced phosphorus compounds in soil and potential impacts of climate change. *European Journal of Soil Science*, 72(6), 2517-2537.

Minor comments

Lines 29-30. "Warming decelerated the gross rates of phosphate mobilization by 21%, likely due to decreased soil total P pools (substrates)" What here means mobilization? Leaching? Release by mineralization?.

3-2. Author response:

Thanks for pointing this out! The gross rates of phosphate mobilization include both biotic (microbial) and abiotic phosphate mobilization (release), i.e., organic P mineralization, and P desorption and dissolution. We now added this information to the revised manuscript, "Warming decelerated the gross rates of phosphate mobilization (including microbial P mineralization, and P desorption and dissolution) by 21%,..." (lines 33-34).

Line 219. “temperate forest soils to become P limited⁵⁹” Moreover this is in the end Results/discussion paragraph that should be a synthesis of this study and the cite referring to other study provide some confusion. Thus I advise to make a concrete reference of the pass studies. For example “Thus warming reduced P-availability..... in this study consistent with previous studies suggesting temperate forest soils under warming can become P limited⁵⁹”

3-3. Author response:

We deleted the reference. Besides, combined with comment 4-19, we revised this sentence to “and can turn the soil microbes in temperate forest soils to become P limited.” (line 255-256).

I advise a minor revision previous to publication

Reviewer #4 (Remarks to the Author):

Tian et al. revised their manuscript following the comments of two reviewers. In the revised manuscript, they (i) extended the method descriptions, (ii) included a new figure and a new table, and (iii) rewrote parts of the manuscript. Altogether, the revision was done very thoroughly and most concerns of the reviewers were resolved. Yet, the introduction and hypotheses need to be improved and a broader perspective need to be implemented still. Because some of the findings challenge textbook knowledge, the authors must put more effort in eliminating alternative explanations. Finally, some minor issues should be incorporated before the study can be published. In the following, the **original reviewer comments are given in orange**, the **authors’ response in green** and my new comments in black.

Comments on author responses to reviewers

Reviewer #1: In its current format the manuscript fails to provide sufficient background to provide a rationale for the presented hypotheses

Author response (shortened): We wrote a separate paragraph to explain why soil depth and seasonality are important in the context of this study.

New reviewer: There is still a mismatch between the background provided in the introduction and the hypotheses. On the one hand, neither soil depth nor seasonality (newly introduced) form part of the hypotheses. On the other hand, the expected directions of the warming effects (“promote”, “increase”, “decrease”) are not backed up by explanations in the preceding introduction. For example, lines 49-54 do not explain responses of P cycling that are later stated as hypotheses.

Therefore, the structure of the introduction and its match with the hypotheses need to be improved.

4-1. Author response:

Thanks for the comment! We revised the introduction, more tightly linking the hypotheses with the preceding introduction (lines 56-67 and 83-89). Regarding soil depth and seasonality, the main focus of this study is long-term soil warming effects on soil P cycling. The reason for including soil depth and seasonality in this study was to provide a more thorough understanding of these warming effects across soil depths and seasons, i.e., to show whether warming effects are consistent across seasons and soil depths or whether they are more context-dependent. Thus, we prefer to keep the story focused and only address the potential interactive soil depth and seasonality effects in the introduction (lines 73-79) and in the discussion (lines 256-259). Formulating hypotheses for seasonal and soil depth effects would also generally be hard, given the lack of literature on such effects on P cycling under warming scenarios.

Revised manuscript:

Soil P cycling is driven by biotic and abiotic processes. Some studies have analyzed the biotic side by investigating warming effects on soil phosphatase activities and on soil P mobilization (Margalef et al., 2021; Rui et al., 2012; Zhou et al., 2021). Globally, warming did not increase phosphatase activity (Margalef et al., 2021), likely as a stimulative warming effect was offset due to concurrently reduced soil water availability (Sardans et al., 2006; Zuccarini et al., 2020). On the other hand, the mineralization of organic P was stimulated (Rui et al., 2012) and soil P mobilization increased with warming (Zhou et al., 2021), accompanied by increased plant P demand (Lie et al., 2022) and uptake (Zhou et al., 2021). Abiotic processes, i.e., adsorption/desorption and precipitation/dissolution processes, affect P availability in soil solution, adding another component to the complexity of soil P cycling. Adsorption is a thermodynamically driven process and phosphate sorption to dolomite has been shown to be an exothermic reaction (Conant et al., 2011; Hanyabui et al., 2020; Yuan et al., 2015). Thus, increasing temperatures are expected to shift the equilibrium to a state of less adsorbed P, but this was not demonstrated yet...

Hypotheses: long-term warming will (i) promote biological soil P processes, i.e., phosphomonoesterase activity due to the lack of drought offsets of the warming effect, (ii) hinder P sorption relative to desorption, and (iii) thereby increase gross rates of soil P mobilization through biotic and abiotic processes. The increase in soil P mobilization will be reflected in increased microbial (and fine root) P uptake, but will increase the risk of soil P losses at the same time.

Newly added reference:

Conant, R. T., Ryan, M. G., Ågren, G. I., Birge, H. E., Davidson, E. A., Eliasson, P. E., ... & Bradford, M. A. (2011). Temperature and soil organic matter decomposition rates—synthesis of current knowledge and a way forward. *Global change biology*, 17(11), 3392-3404.

Yuan, X., Xia, W., An, J., Yin, J., Zhou, X., & Yang, W. (2015). Kinetic and thermodynamic studies on the phosphate adsorption removal by dolomite mineral. *Journal of Chemistry*, 2015.

Reviewer #1: Lines 205-222: The conclusions paragraph is another example of the lack of depth here. The paragraph reads as a list of results and includes no references to other work.

Author response: We shortened the results and modified the conclusion. In a conclusion, however, it is not usual to cite (multiple) references, but rather to summarize the results and give an outlook on the implications of the work to be published. We therefore remarked the importance of this study: 1) providing novel data of long-term warming effects on soil P process rates; 2) long-term warming can drive temperate forest soils to become P limited; 3) which can further affect soil C and N cycling.

New reviewer: In addition to my comment on the phrasing (see below), I am still not convinced by the broader context of the conclusion. Why should we worry about soil (micro)organisms whose growth is potentially limited by P? And in which way would this be linked to soil C and N cycling – again stressing the relevance of this link? Could P-limited soil (micro)organisms have an impact on the C sink function of the soil? Could P-limited soil (micro)organisms influence N losses (open N cycle if N is no longer limiting)? I strongly recommend that the authors spend some more thoughts on the broader implications of their study.

4-2. Author response:

We revised conclusion and now specified how microbial P limitation would potentially affect ecosystem C and N cycling (lines 250-274).

Revised manuscript:

In conclusion, this study contributes novel data on warming effects on soil P processes, providing a more comprehensive understanding of climate change effects on the complex soil P cycle (Fig. 1). Our results suggest that long-term warming reduced bioavailable P, which resulted from substantial losses of TP (likely via plant P uptake and downward P transportation) and increased P_i sorption (onto Fe oxyhydroxides and clay minerals), and can turn the soil microbes in temperate forest soils to become P limited. Moreover, according to the results of the mixed-effects models, most of the measured P pools and processes showed no interactions between warming and soil depth and/or season (Table 2), indicating consistent effects of long-term soil warming on the P cycle across different soil depths and seasons. In general, forest ecosystems on limestone and dolomite are P limited (Prietzl et al., 2014; Prietzl et al., 2022), due to the low P content of bedrock. Losses of P in these forests happen continuously by weathering, erosion and subsurface transport, and tree P demand is largely met via internal recycling, indicated by the upregulation of phosphatases and the shift from microbial P to plant fine root P pools under P limitation. The data therefore show that P constraints are likely amplified for both, microbes and plants, under warming. Given the strong signs that European forests have become increasingly P limited during the last decades (Talkner et al., 2015; Du et al., 2021), we therefore expect that the intensified competition between plants and soil microbes for this scarce resource will have negative consequences for plant net primary production and ecosystem C sequestration with climate warming. We however note that extrapolations of the soil warming effects found here to the ecosystem level needs to be done cautiously, given that we only warmed soils, and here only parts of a trees' root system, but not the aboveground compartment. Whole ecosystem warming might even further impair the delicate balance between P limited trees and soil microbes, given that warming increases the plant P demand.

Newly added references:

Prietzl, J., Christophel, D., Traub, C., Kolb, E., & Schubert, A. (2015). Regional and site-related patterns of soil nitrogen, phosphorus, and potassium stocks and Norway spruce nutrition in mountain forests of the Bavarian Alps. *Plant and soil*, 386(1), 151-169.

Prietzl, J., Krüger, J., Kaiser, K., Amelung, W., Bauke, S. L., Dippold, M. A., ... & Lang, F. (2022). Soil phosphorus status and P nutrition strategies of European beech forests on carbonate compared to silicate parent material. *Biogeochemistry*, 158(1), 39-72.

Du, E., van Doorn, M., & de Vries, W. (2021). Spatially divergent trends of nitrogen versus phosphorus limitation across European forests. *Science of the Total Environment*, 771, 145391.

Talkner, U., Meiwes, K. J., Potočić, N., Seletković, I., Cools, N., De Vos, B., & Rautio, P. (2015). Phosphorus nutrition of beech (*Fagus sylvatica* L.) is decreasing in Europe. *Annals of forest science*, 72(7), 919-928.

Reviewer #2: The authors argue possible biotic pathways to explain the decrease of total P pools. This includes a higher uptake by plants explained by an increase in fine root production. What are the implications and limitations of the experimental design to evaluate plant response to warming? In contrast to what happens with current climate change, in this case the soil is warmed but not the aboveground which restricts or unbalances plant response to belowground adaptations.

Author response: We agree with the reviewer that the results from soil warming experiment and soil+canopy warming experiment may differ, however, due to technical and financial constraints we are not aware of any warming experiment of a full mature temperate, boreal or tropical forest (above+belowground) with an average canopy height of >25 m. We would have ideas to do so, but this would also incur very strong unwanted micrometeorological effects and cost multi-millions of Euros to setup. With canopy warming, it most likely can increase gross plant P uptake from soil due to facilitated photosynthesis, but can also increase litter P inputs to soils. However, due to the relocation of foliar P from old leaves to other tissues, especially in P restricted environments, soil+canopy warming experiment may

have higher net plant P uptake than soil warming experiment. We are fully aware of that we only warmed limited part of trees in each plot. This was why we toned down the interpretation of plant P uptake pathway in our manuscript (Lines 102-105 and Lines 118-121).

New reviewer: I fully agree with the arguments of the authors. However, this constraint needs to be discussed briefly in the main text of the manuscript. If this explanation is missing, the reader will be tempted to infer implications e.g., on growth limitation of the forest by P and its consequences for the forest C sink, that must not be drawn based on the experimental design.

4-3. Author response:

We now addressed that our experiment is based on soil warming (without canopy warming) in the manuscript. We amended the text in the Discussion “However, due to the small plot size (2 by 2 meters) and soil-only warming (without canopy warming), the roots and canopies of trees are shared across the warming and the control treatments, and across experimental and non-experimental areas, causing comparable litterfall P returns in warmed and un-warmed areas. Therefore, increased gross plant P uptake in the warming treatment combined with comparable plant litter P returns between treatments will result in the redistribution of P, ultimately reducing soil TP in warmed soils in the long run.” (lines 107-114)

To make this point clearer we further added in the Conclusions that: “We however note that extrapolations of the soil warming effects found here to the ecosystem level needs to be done cautiously, given that we only warmed soils, and here only parts of a trees’ root system, but not the aboveground compartment. Whole ecosystem warming might even further impair the delicate balance between P limited trees and soil microbes, given that warming increases the plant P demand.” (lines 269-274)

Reviewer #2: Higher Fe hydroxide content after warming treatments is attributed to faster weathering processes. This affirmation is surprising as a result of a 14-years warming. This is because weathering intensity directly relates to temperature, CO₂ concentration and precipitation but it is generally attributed to longer time scales (millennial). There aren’t many published studies considering the role of weathering changes at decadal time scales. I would appreciate that the authors develop more this idea including more references. At the moment the effects of the climate change induced changes in weathering are not well studied and understood.

Author response: We agree with the reviewer that normally we would expect changes in soil metal oxide contents (as well as soil texture and mineralogy) would happen over longer time periods, and there are not many warming studies that have investigated this topic, given the common belief that soil texture and mineralogy does not change within 10-20 years. However, this is a belief and not grounded in science, and we collated all literature and cited it showing that global change (land use, elevated CO₂, warming) can change this at decadal scales. We were also surprised, but we show that this change happened. So this is not speculation, but grounded on robust data. We now wrote a separate paragraph, discussing the warming effects on soil metal oxides, soil texture and clay mineralogy. Due to the scarcity of other studies, we cited several examples of decadal changes of soil metal oxides, soil texture and mineralogy to soil management activities to strengthen the discussion. Besides, we will continue researching on this topic at our site in the future.

New reviewer: With due respect, the time scales associated with weathering are no belief but itself also corroborated by data (see body of literature on changes in soil properties in chronosequence studies). Nevertheless, I am in favor of challenging the long-term perspective of weathering. However, this can only be achieved by thoroughly eliminating alternative explanations for your findings. One such explanation is an a priori difference between the pairs of plots. For example, the high heterogeneity within the control and warming treatments (e.g. l. 238-239) increases the risk of initially selecting control and warming plots differing in soil properties – just by chance. These concerns can be dispelled by data on soil properties of

the plots at the time of the experimental set up i.e., before the start of the warming treatment. Are data on initial soil properties such as soil texture and mineralogy available? Or do archived soil samples exist that could be analyzed still? How do these data relate to your findings 14 years later? Without data on initial soil properties, the causality of the warming effects on soil texture and mineralogy remains questionable and all corresponding statements need to be toned down.

4-4. Author response:

Thanks for pointing this out. You are right and we therefore ask to apologize for overdoing our response. The experimental set-up meant that across the site at six locations paired plots were established directly aside of each other, both manipulated by burying heating cables, but only one set functional (heated) and the other unfunctional (control). This block design controls for much of the spatial heterogeneity one finds in mountain forests. During data analysis, we took the block effects into consideration and thus we set block as random factor in the mixed effects models. Unfortunately, we do not have historical data of the original soil conditions, neither do we have archived soils to compared historical plot data to those measured now. Therefore, we toned down the corresponding statements (lines 188-198).

Revised manuscript

Short-term (decadal) warming effects on soil texture and mineralogy have rarely been studied. Elevated temperatures favor soil weathering processes (Neupane et al., 2021; Qafoku 2015), potentially increasing soil metal oxide contents at long time scales. Besides, higher temperatures can increase the soil clay contents (Buol et al., 1990; Jenny 1994). However, these warming-induced changes are commonly observed at longer time scales, e.g. at centennial or millennial time periods (Qafoku 2015; Buol et al., 1990; Jenny 1994). Comparatively rapid (decadal) changes in soil metal oxides, soil texture and/or clay mineralogy have been shown in a glacier chronosequence study (Bernasconi et al., 2011) and in other studies in response to changes in management and fertilizer regime (Qafoku 2015; Cornu et al., 2012; De Mastro et al., 2020; Liu et al., 2017; Fink et al., 2014). If such rapid changes are triggered by soil warming as well, forest nutrient cycling can be negatively affected by increased sorptive constraints on substrate availability of these processes in the near future.

Newly added references:

Bernasconi, S. M., Bauder, A., Bourdon, B., Brunner, I., Bünemann, E., Chris, I., ... & Zumsteg, A. (2011). Chemical and biological gradients along the Damma glacier soil chronosequence, Switzerland. *Vadose Zone Journal*, 10(3), 867-883.

Reviewer #2: Apart from clay mineralogy, what are the primary minerals? Which is the mineral phase attributed to the crystalline Iron? When compared with the total mineralogy (not only clays), is it possible to estimate the % of amorphous and crystalline Iron?

Author response: The primary mineral is dolomite at our site. Goethite (iron hydroxide; α -FeO(OH)) constitutes the main phase of crystalline Fe in all soil samples at the Achenkirch site. We determined the content of crystalline iron and of amorphous iron in soils by wet chemistry (dithionite- and oxalate-extractable Fe). Oxalate-extractable represents the amorphous Fe oxyhydroxides (plus some humic-bound Fe) and dithionite-extractable minus oxalate-extractable Fe is the crystalline Fe. For the determination of soil mineralogy, we used X-ray diffraction (XRD), which only detects crystalline phases such as goethite, lepidocrocite, pyrite, hematite, etc. Ferrihydrite is mostly amorphous or very poorly crystallized. It is an Fe mineral, which can hardly be detected by XRD. Based on the results of Fe oxide contents, we get a range for the crystallised iron mineral (goethite) from 0.2 to 0.6 mass%, and a range for the amorphous iron (ferrihydrite) from 0.07 mass% to 0.19 mass%. These results also fit quite well to the results by XRD, where we found goethite in traces, which means <1 mass%.

New reviewer: The authors should include this description in the methods' part of the manuscript.

4-5. **Author response:**

We now added "The primary soil minerals are quartz, plagioclase, dolomite and K-feldspar at the study site (Supplementary Fig. 1). Moreover, goethite (iron hydroxide; α -FeO(OH)) constitutes the main phase of crystalline Fe, ranging from 0.2 to 0.6 mass%, and amorphous iron (ferrihydrite) ranges from 0.07 to 0.19 mass%." in the paragraph where we introduce the characteristic features of our site in the Methods section (lines 301-304). In terms of the methods for determination of metal oxyhydroxides and clay mineralogy, these were already detailed in the former manuscript (lines 379-403).

Additional comments by new reviewer:

36-37 The conclusive sentence is too broad. Link back to climate change, what would this mean for the forest? P limitation = detrimental effects on growth?

4-6. **Author response:**

We revised the conclusive sentence to "This study therefore highlights how long-term soil warming triggers changes in biotic and abiotic soil P pools and processes, which can potentially aggravate the P constraints of the trees and soil microbes and thereby negatively affect the C sequestration potential of these forests." (lines 41-42).

45-48 and 54-65 redundant, please merge.

4-7. **Author response:**

We rewrote this part of the introduction section, therefore circumventing this repetition (lines 49-55).

80 no mention of references within hypotheses (should be explained already in the introduction)

4-8. **Author response:**

We removed the reference from the hypothesis and added it to the corresponding introduction section (lines 64-67).

98 "content" is ill defined. Do you refer to stocks or concentrations?

4-9. **Author response:**

We replaced "content" with "concentration" (line 102).

109-110 Unclear: does "dissolved P in soil water" refer to DIP, DOP or TDP concentrations?

4-10. **Author response:**

We revised to "dissolved inorganic P in soil water" (line 119).

115, 119 Awkward phrase “leaching of Olsen-Po”: Here you interpret your results and thus, go beyond your own measurements. Replace by “leaching of Po” (or DOP if you consider the dissolved phase most relevant)

4-11. Author response:

Based on comment 3-1, we revised the corresponding paragraph. This comment is no longer applicable.

118-121 Statements too strong because you did neither determine DOP leaching nor root P uptake/reflux, please tone down. “net losses of total P from the soils seem to be mainly due to” and “as the increased root P uptake likely was largely compensated”.

4-12. Author response:

Based on comment 3-1, we revised the corresponding paragraph. This comment is no longer applicable.

173-174 Where are these results shown/specified?

4-13. Author response:

We added the X-ray diffractograms as Supplementary Fig. 1.

177 Statement too strong, please rephrase (“it cannot explain the higher abiotic P immobilization”).

4-14. Author response:

We revised to “it does not explain the higher abiotic P immobilization.” (lines 207-208).

189 “stronger stimulation” (because “than” follows in the second part of the sentence)

4-15. Author response:

Done. Thanks for reminding us of this (line 224).

217 “(likely via leaching and plant P uptake)”

4-16. Author response:

Due to the comment 3-1, we needed to revise this sentence, and we also integrated your suggestion to tone down the sentence (lines 253-254).

198/200/218 “Fe oxide” has not been introduced. Up to here (l. 198), you referred to Fe oxyhydroxides/dithionite-/oxalate-extractable Fe. Please improve terminology and the explanations on the different fractions when mentioned first.

4-17. Author response:

We replaced “Fe oxide” with “dithionite extractable Fe” or “Fe oxyhydroxides” throughout the manuscript to be more accurate (lines 234, 235, and 254).

216-217 Awkward phrase “soil available P”. Commonly, terms such as “plant-available/bioavailable P” in soil are used. Please rephrase.

4-18. **Author response:**

We replaced “soil available P” with “bioavailable P” (lines 172, 229, and 253).

218-219 Awkward phrase “can turn temperate forest soils to P limited”. By definition, the growth of organisms – but not the soil – can be limited by nutrients. Please rephrase.

4-19. **Author response:**

We changed the sentence to “can turn the soil microbes in temperate forest soils to become P limited” (lines 255-256).

234 Add proper citation for soil classification!

4-20. **Author response:**

We added new citation for soil classification (line 286).

Newly added reference:

IUSS Working Group WRB. 2015. World Reference Base For Soil Resources 2014, Update 2015 International Soil Classification System For Naming Soils And Creating Legends For Soil Maps. World Soil Resources Reports No. 106 (FAO, Rome).

244 “The high contents of kaolinite”

4-21. **Author response:**

Done. Thank you! (line 296)

291-301 This description is confusing, for Olsen P just NaHCO₃ is used as extractant. However, the authors state that they used persulfate digestion in addition (l. 291-292). Please clarify how the Olsen P method was embedded (?) in the procedure used for the determination of MBP.

4-22. **Author response:**

We used the persulfate digestion method to convert dissolved organic P (DOP) to DIP, allowing to measure total dissolved P (TDP) in both, fumigated and non-fumigated soils after bicarbonate extraction. TDP was used to calculate DOP (non-fumigated soils) and microbial biomass P from TDP in fumigated minus non-fumigated soils. Then, we used the malachite green method for determination of phosphate contents. We hereby can determine:

- Olsen P_i: from untreated soil samples
- Olsen P_t: from acid persulfate digested, non-CFE soil samples
- Olsen P_o: Olsen P_t minus Olsen P_i
- MBP: acid persulfate digested, CFE soil samples minus acid persulfate digested non-CFE soil samples with a correction factor.

We revised the description to make this clearer (lines 346-360).

Revised manuscript:

Olsen total P (Olsen P_t), Olsen inorganic P (Olsen P_i), Olsen organic P (Olsen P_o), and MBP were determined using soil bicarbonate extraction and the malachite green method (D'Angelo et al., 2001) with/-out previous acid persulfate digestion (Rowland et al., 1997) and chloroform fumigation-extraction (CFE) (Vance et al., 1987). Briefly, we weighed two aliquots of each soil sample (2 g fresh weight) into suitable sample containers. The first aliquots were fumigated with chloroform for 48 hours, the second ones served as non-fumigated controls. Then, all soil samples were extracted with 0.5 M NaHCO_3 (pH=8.5, 1:7.5 (w:v)) and filtered. All samples were acidified by addition of 2.75 M H_2SO_4 (10% of the extract volume), and subsamples of these extracts were digested by acid persulfate reagent (0.185 M sodium persulfate in 0.5 M H_2SO_4) with a ratio of extract: reagent of 20:3. Reactive inorganic PO_4^{3-} of all samples was measured by the malachite green method (D'Angelo et al., 2001). Olsen P_t and Olsen P_i were calculated from the results of non-CFE digested and undigested aliquots, respectively. Olsen P_o equals the difference between Olsen P_t and Olsen P_i , while MBP was determined as the difference between digested CFE and digested non-CFE aliquots, applying a correction factor K_{EP} of 0.4 (Brookes et al., 1982).

332 Why Fe oxides? There are also crystalline Fe oxyhydroxides.

4-23. Author response:

We changed to "crystalline Fe oxides and oxyhydroxides" (lines 388-389).

Table 1:

- Please add the number of replicates ($n = 6$ for each treatment) in the caption. However, I do not get why some measurements total 18 while others sum up to 6 (see footnote). Were pseudoreplicates used in the former case?
- What does d.s. mean? Dry soil? Please explain in caption.
- Add different letters (a, b) for all cases in which the control and the warming plots differed significantly.

4-24. Author response:

We added the number of replicates (sample size), explained the abbreviation of d.s., and added different letters, to indicate whether there was a significant difference among treatments, in the caption of Table 1. Regarding the number of replicates, the parameters that have 18 replicates are because we measured these parameters from 6 blocks in 3 seasons in 2019, whereas the parameters, which have 6 replicates were determined once from soil samples in 2020. We described our sampling scheme in the Methods section (lines 319-330).

Revised manuscript:

Data presented are means \pm standard error. Different letters indicate significant differences among treatments at two soil depths according to ANOVA followed by Tukey test. Soil organic C, microbial biomass C, N, and P, sand content, and exchangeable Na^+ , K^+ , and Mn^{2+} were log-transformed to meet the assumptions prior to run the Tukey test. The sample size of pH, soil water content, soil organic C, total soil N, total soil P, and microbial biomass data is 18 (six blocks sampled in three seasons). The sample size of soil metal oxyhydroxide contents, soil texture, and exchangeable cations is 6 (six blocks). Abbreviations: d.s. refers to dry soil.

Figure 1: You did not measure plant P (= either above- or belowground P stocks or both). Stick to those variables for which results are presented in this manuscript.

4-25. **Author response:**

The aim of this schematic illustration is to visualize the P flows at our site. Since plant P uptake is an important pathway explaining soil P losses and of plant responses to P limitations, we would still like to keep it in the schematic illustration. We now added the reference number next to plant P uptake in the figure and explained in the caption that these results are retrieved from Kengdo et al., 2022, who studied fine root production and element content at the same site in the same year.

Reference:

Kwachko Kengdo, S., Peršoh, D., Schindlbacher, A., Heinzle, J., Tian, Y., Wanek, W., & Borken, W. (2022). Long-term soil warming alters fine root dynamics and morphology, and their ectomycorrhizal fungal community in a temperate forest soil. *Global Change Biology*, 28(10), 3441-3458.

Fig. 3: The calculation of crystalline Fe includes dithionite extracted P. Therefore, show only one (delete either subfigures a+b or c+d).

4-26. **Author response:**

Thank you for the suggestion. Since dithionite extracted Fe includes both crystalline and amorphous Fe oxyhydroxides, we decided to keep sub-figures a+b and remove sub-figures c+d (for the updated figure, please see the revised display document).

REVIEWERS' COMMENTS

Reviewer #3 (Remarks to the Author):

The manuscript "Long-term soil warming decreases microbial phosphorus utilization by increasing abiotic phosphorus sorption and phosphorus losses" by Tian et al. take profit of a long time manipulation of warming in Alpine forests to study the long term impacts on soil P status. The experiment is correct; consist in a block design with six blocks with a control and warming treatment. The study must be considered seriously to be published by the solidity of the experimental design, the longtime run of the warming treatment and overall of the lack of knowledge of warming impact of "in situ" P status. I have observed that the current version is the result as a first revision; after reading detailed the manuscript without doubt, the manuscript needs many improvements/corrections/clarifications to clearly state and identify the take home lessons that it has. Thus, my advice is to return the manuscript to the authors to improve it previously to its publication. See my detailed comments.

Comments for Authors

1. I advise to move the sentence of lines 35-36 before the sentence in lines 33-35.
2. Lines 38-39 "... , though this did not.....microbial biomass P" what exactly this means, what have exactly of different microbial biomass P from biotic P, immobilization, at least both overlap!!!
3. Abstract in general, the overall results allows to make a general picture of the results, something like " Warming increased the sorption to more recalcitrant soil P fractions (absorbed in oxyhydroxydes and clays), and higher outputs of P from soils by increasing plant P uptake and vertical fluxes of colloidal and particulate P contributing all them to the drop of available and labile soil P fractions in warmed soils. Despite microbes increased acid phosphatase activity this was not sufficient to avoid a decrease of available and labile soil P forms." Take note that the title ends with "...phosphorus losses" thus this question should be clearly stated also in the abstract.
4. Line 51 "... , showing overall increases in plant N:P ratios" Please add some bibliography support to sustain this part of the sentence.
5. Lines 57-58. Please clarify "...activities and on soil P mobilization6,14,15." Soil mobilization in what aspect(s), leaching, litter and SOC mineralization, leaching from minerals, sorption-desorption,..?
6. Lines 70-71, to what concrete process of mobilization and immobilization are you referring to?
7. Line 91 change "losses" to "decreases"
8. Lines 96-97. Was this sentence referred to the experimental site?
9. Lines 99-114. It is not clear if all this sentences are referring to the Kengdo et al study or at least in part to the present study. Please clarify.
10. Very important part. Lines 123-134. Have authors some further result sustaining the soil pores changes favoring (or potentially favoring) the seepage of P?
11. Lines 142-145. Phosphines are negligible in nature , several reports conclude it I advise to take out this from the manuscript.
12. Lines 147-150. I suggest to clearly stated this sentence to provide a clear take home lesson of the concrete results of the current study, I suggest: "In our experiment the results strongly suggest that the main causes of total soil P losses unde warming are due to an increase of plant P-uptake and subsurface dissolved and colloidal/particulate P fluxes.
13. Lines 156-157 Change to more clear: "This formation of more recalcitrant P stocks could have taken place via sorption (on to....., forming Ca-apatite (Fig. 1)."
14. Lines 176-177. Please take into account that Kaolinites are 1:1 clays with a low capacity to absorb.
15. Lines 185-187. Yes more clay and less sand should increase at some extend the capacity to absorb P in this case increase it, but moreover also should make more difficult the P infiltration, counteracting the effect of more porosity by higher root turn-over. This together with the fact that if we argue that plants taken up more P this should be accompanied by more growth and water up-take under warming decreasing the potential of nutrient soil mobilization by infiltration. This is the main not clear/resolved variable, that should be specifically clarified by authors, putting altogether the variables that can play in the increase or decrease of P infiltration and loss from the studied upper parts of soil.

16. Line 199 contents or concentrations?

17-Lines 221-229. But the experiment of Shi et al (now under revision) was conducted in the same experimental site that the current experiment?

18. Line 253. "substantial losses of soil TP"

19. Line 254. "increased Pi sorption and accumulation of P in soil recalcitrant fractions (e.i. onto Fe oxyhydroxides..."

20. In general in the last conclusions paragraph. State clearly if plants grow more under warming, in this case warming allows plant to accumulate sources and take profit in production capacity with a net transfer of P from soil to plant, being the microbes the losers in the competition plant-microbes under warming, a very important question that should be clarified here.

21. Line 296. The authors should check the use of concentrations and contents throughout the manuscript.

Reviewer #4 (Remarks to the Author):

Tian et al. revised their manuscript a second time following the comments of two reviewers. In the revised manuscript, they streamlined the introduction and discussion, they embedded their study in a broader context and they implemented all other changes requested by the reviewers. I would like to congratulate the authors for their careful and comprehensive second revision. There is only one tiny issue left that has not been fully addressed. Without repeating reviewer comments and author responses, I would like to draw the attention to the mentioning of soil depth and seasonality in the introduction (lines 73-79). I agree with the authors that the story should be kept focused and soil depth/seasonality should not be included in the hypotheses. Following this reasoning, these two aspects do not need to be explained in the introduction where they raise misleading expectations. Therefore, I recommend to delete this paragraph in the introduction and just use it to highlight the robustness of the findings in the discussion.

We thank the reviewers for their comments, which helped us improve the manuscript. We addressed each comment individually. **The responses are in green** for clarity, and corresponding changes are **highlighted in yellow** in the revised manuscript.

REVIEWERS' COMMENTS

Reviewer #3 (Remarks to the Author):

The manuscript "Long-term soil warming decreases microbial phosphorus utilization by increasing abiotic phosphorus sorption and phosphorus losses" by Tian et al. take profit of a long time manipulation of warming in Alpine forests to study the long term impacts on soil P status. The experiment is correct; consist in a block design with six blocks with a control and warming treatment. The study must be considered seriously to be published by the solidity of the experimental design, the longtime run of the warming treatment and overall of the lack of knowledge of warming impact of "in situ" P status. I have observed that the current version is the result as a first revision; after reading detailed the manuscript without doubt, the manuscript needs many improvements/corrections/clarifications to clearly state and identify the take home lessons that it has. Thus, my advice is to return the manuscript to the authors to improve it previously to its publication. See my detailed comments.

Comments for Authors

1.I advise to move the sentence of lines 35-36 before the sentence in lines 33-35.

3-1. Author Response:

Combined with the next two comments, we revised our Abstract as follow, "Phosphorus (P) is an essential and often limiting element that could play a crucial role in terrestrial ecosystem responses to climate warming. However, it has yet remained unclear how different P cycling processes are affected by warming. Here we investigated the response of soil P pools and P cycling processes in a mountain forest after 14 years of soil warming (+4°C). Long-term warming decreased soil total P pools, likely due to higher outputs of P from soils by increasing net plant P uptake and downward transportation of colloidal and particulate P. Warming increased the sorption strength to more recalcitrant soil P fractions (absorbed to iron oxyhydroxides and clays), thereby further reducing bioavailable P in soil solution. As a response, soil microbes enhanced the production of acid phosphatase, though this was not sufficient to avoid decreases of soil bioavailable P and microbial biomass P (and biotic phosphate immobilization). This study therefore highlights how long-term soil warming triggers changes in biotic and abiotic soil P pools and processes, which can potentially aggravate the P constraints of the trees and soil microbes and thereby negatively affect the C sequestration potential of these forests." (Lines 28-42)

2.Lines 38-39 "..., though this did not.....microbial biomass P" what exactly this means, what have exactly of different microbial biomass P from biotic P, immobilization, at least both overlap!!!

3-2. Author Response:

Indeed, biotic P immobilization (here the process of microbial $^{33}\text{P}_i$ uptake) and microbial biomass P (a soil P pool, not a process, driven by the balance of P uptake, storage and turnover/death, but certainly larger biomass also triggers greater microbial P uptake) are two parameters that are or can be highly connected. Since we measured both parameters with different methods, we meant to present both results, which are consistent with each other. Please see the revision in response 3-1.

3.Abstract in general, the overall results allows to make a general picture of the results, something like " Warming increased the sorption to more recalcitrant soil P fractions (absorbed in oxyhydroxydes and clays), and higher outputs of P from soils by increasing plant P uptake and vertical fluxes of colloidal and particulate P contributing all them to the drop of available and labile soil P fractions in warmed soils. Despite microbes increased acid phosphatase activity this was not sufficient to avoid a decrease of available and labile soil P forms." Take note that the title ends with "...phosphorus losses" thus this question should be clearly stated also in the abstract.

3-3. Author Response:

Thanks for your great suggestion and your synthesis of this study. We revised our Abstract accordingly. Please see response 3-1.

4.Line 51 "..., showing overall increases in plant N:P ratios" Please add some bibliography support to sustain this part of the sentence.

3-4. Author Response:

Reference added. Thanks.

Added Reference:

Yue, K. et al. Effects of three global change drivers on terrestrial C: N: P stoichiometry: a global synthesis. *Glob. Chang. Biol.* **23(6)**, 2450-2463 (2017).

5.Lines 57-58. Please clarify "...activities and on soil P mobilization6,14,15." Soil mobilization in what aspect(s), leaching, litter and SOC mineralization, leaching from minerals, sorption-desorption,..?

3-5. Author Response:

We now specified that the soil P mobilization here refers to soil organic P mineralization (Line 58).

6.Lines 70-71, to what concrete process of mobilization and immobilization are you referring to?

3-6. Author Response:

We re-wrote the sentence to make it clearer. "... we here applied a ³³P isotope pool dilution method (Wanek et al.,2019), which allows to quantify the process rates of gross and net (biotic and abiotic) P mobilization (reflecting microbial P mineralization as biotic process, and P desorption and dissolution as abiotic processes) and P immobilization (reflecting microbial P immobilization, and P sorption and precipitation)." (Lines 69-73)

7.Line 91 change "losses" to "decreases"

3-7. Author Response:

Done.

8.Lines 96-97. Was this sentence referred to the experimental site?

3-8. Author Response:

No, this sentence refers to the general situation in temperate forest ecosystems (and this also applies to our site, i.e., increased fine root biomass and unchanged fine root C:P ratio). We now revised the sentence to make it clearer. "In temperate forest ecosystems, elevated temperatures generally facilitate plant growth while warming globally had no effects on plant C:P ratios (Yue et al., 2017; Lu et al., 2013)." (Lines 94-95)

9.Lines 99-114. It is not clear if all this sentences are referring to the Kengdo et al study or at least in part to the present study. Please clarify.

3-9. Author Response:

We only referred to the Kengdo et al. study from line 97 to line 103. From line 105 onwards, we discuss the experimental design of the Achenkirch project and its impact in general, which applies to both Kengdo et al. and this study. We now specified this in lines 105-106.

10.Very important part. Lines 123-134. Have authors some further result sustaining the soil pores changes favoring (or potentially favoring) the seepage of P?

3-10. Author Response:

Here we stated that increased fine root growth and turnover may cause the formation of preferential flow paths in soils. We unfortunately do not have further data on the pore size distribution and structure of the control and warmed soils (such as gained by e.g. μ CT – micro computed tomography - or soil water potential curves). Seepage was not quantified at individual plot level, but only subsampled for chemical analyses. Hence, we toned down the sentence, “Hereby, high precipitation at this forest site and the potential increase in root-induced preferential flow paths in warmed soils could have favored dissolved and colloid-associated P to migrate to deep soil layers, causing enhanced soil P losses in the warming treatment (Sohrt et al., 2017; Missong et al., 2018)”. (Lines 129-132)

11.Lines 142-145. Phosphines are negligible in nature , several reports conclude it I advise to take out this from thhe manuscript.

3-11. Author Response:

Removed (also made corresponding revision in Fig. 1).

12.Lines 147-150. I suggest to clearly stated this sentence to provide a clarer take home lesson of the concrete results of the current study, I suggest: “In our experiment the results strongly suggest that the main causes of total soil P losses unde warming are due to an increase of plant P-uptake and subsurface dissolved and colloidal/particulate P fluxes.

3-12. Author Response:

We now made the sentence more direct. “Overall, (greater) net losses of TP from the warmed soils mainly resulted from increased net plant P uptake and subsurface dissolved and colloidal/particulate P fluxes.” (Lines 142-144)

13.Lines 156-157 Cahnge to more clear: “This formation of more recalcitrant P stocks could have taken place via sorption (on to....., forming Ca-apatite (Fig. 1).”

3-13. Author Response:

Done. Thanks!

14. Lines 176-177. Please take into account that Kaolinites are 1:1 clays with a low capacity to absorb.

3-14. Author Response:

We revised the sentences as follow, “In this study, clay mineralogy did not change and was dominated by moderate P-sorbing kaolinites (~60%), chlorites associated with a mixed layered clay (~30%), and illites (~10%).” (Lines 171-173)

15. Lines 185-187. Yes more clay and less sand should increase at some extend the capacity to absorp P in this case increase it, but moreover also should make more difficult the P infiltration, counteracting the effect of more porosity by higher root turn-over. This together with the fact that if we argue that plants taken up more P this should be accompanied by more growth and water up-take under warming decreasing the potential of nutrient soil mobilization by infiltration. This is the main not clear/resolved variable, that should be specifically clarified by authors, putting altogether the variables that can play in the increase or decrease of P infiltration and loss from the studied upper parts of soil.

3-15. Author Response:

Thanks for pointing this out. Indeed, more clay should increase the capacity of P sorption. However, clay (plus Fe oxyhydroxides) associated P can be transferred to deeper soil layers in a substantial amount through the preferential flow pathways (instead of soil matrix flow, which largely transports soluble P forms). We addressed this in lines 206-210.

Regarding the consideration of decreased soil water due to higher plant water uptake, the real-time soil moisture was continuously measured using ECH₂O-10 soil moisture probes (Decagon, Washington, DC, USA) and showed no difference between warming and control treatments (Heinze et al., 2023). Thus, although faster root growth and slightly greater root biomass should allow greater plant water uptake and thereby decrease soil water content, owing to the high mean annual precipitation (~1500 mm), it should have only minimal impact on soil hydrology and therefore not impair downward P transportation.

Newly Mentioned Reference:

Heinzle, J. et al. Soil CH₄ and N₂O response diminishes during decadal soil warming in a temperate mountain forest. *Agric. For. Meteorol.* **329**, 109287 (2023).

16. Line 199 contents or concentrations?

3-16. Author Response:

The unit of exchangeable Ca²⁺ and Mg²⁺ is mmol cation kg d.s.⁻¹, so it should be content.

17-Lines 221-229. But the experiment of Shi et al (now under revision) was conducted in the same experimental site that the current experiment?

3-17. Author Response:

Yes, the experiment of Shi et al., 2023 (now accepted for publication by Global Change Biology, reference added) was conducted at the same experimental site, and they used the same soil samples as ours from the August 2019 sampling. We stated that "Shi et al. (2023) measured the stimulation of microbial growth in response to substrate amendments as an indicator of microbial element limitation **in the same soils in August 2019.**" (Lines 216-218)

Newly Added Reference:

Shi, C. et al. Does long-term soil warming affect microbial element limitation? A test by short-term assays of microbial growth responses to labile C, N and P additions. *Glob. Chang. Biol.* Accepted Author Manuscript (2023). <https://doi.org/10.1111/gcb.16591>

18. Line 253. "substantial losses of soil TP"

3-18. Author Response:

Done. Thanks!

19. Line 254. "increased Pi sorption and accumulation of P in soil recalcitrant fractions (e.i. onto Fe oxyhydroxides..."

3-19. Author Response:

Done. Thanks!

20. In general in the last conclusions paragraph. State clearly if plants growth more under warming, in this case warming allows plant to accumulate sources and take profit in production capacity with a net transfer of P from soil to plant, being the microbes the losers in the competition plant-microbes under warming, a very important question that should be clarified here.

3-20. Author Response:

Thanks for the suggestion. We stated in the Conclusion that "Losses of P in these forests happen continuously by weathering, erosion and subsurface transport, and tree P demand is largely met via internal recycling, indicated by the upregulation of phosphatases and **the shift from microbial P to plant fine root P pools under P limitation.**" (Lines 256-259). We prefer to avoid direct comparison of P uptake between plant and soil microbes, because the plant P data is restricted to fine roots from our partner study (Kendgo et al., 2021), which is not in the dataset of this study. Moreover, soil warming only reached part of the fine root systems of a tree and therefore whole tree responses and tree P uptake cannot be derived from this manipulative study.

21. Line 296. The authors should check the use of concentrations and contents throughout the manuscript.

3-21. Author Response:

We checked the manuscript thoroughly and made changes in lines 100, 412, and 421.

Reviewer #4 (Remarks to the Author):

Tian et al. revised their manuscript a second time following the comments of two reviewers. In the revised manuscript, they streamlined the introduction and discussion, they embedded their study in a broader context and they implemented all other changes requested by the reviewers. I would like to congratulate the authors for their careful and comprehensive second revision. There is only one tiny issue left that has not been fully addressed. Without repeating reviewer comments and author responses, I would like to draw the attention to the mentioning of soil depth and seasonality in the introduction (lines 73-79). I agree with the authors that the story should be kept focused and soil depth/seasonality should not be included in the hypotheses. Following this reasoning, these two aspects do not need to be explained in the introduction where they raise misleading expectations. Therefore, I recommend to delete this paragraph in the introduction and just use it to highlight the robustness of the findings in the discussion.

4-1. Author Response:

Thank you for your positive feedback!

Regarding the comment, we deleted the paragraph, which introduces soil depth and seasonality, and added one sentence, "Moreover, the expected warming effects on P pools and cycling processes were tested across soil depth and season to evaluate their consistency or their context dependence.", to briefly address our experimental setup in lines 85-87.